# The Characteristics of Cloud Macro Parameters Caused by Seeder-Feeder inside Clouds Measured by Millimeter-wave Cloud Radar in Xi'an, China

## Huige Di[*], Yun Yuan

*School of Mechanical and Precision Instrument Engineering, Xi'an University of Technology, Xi'an 710048, China*

* Corresponding author: dihuige@xaut.edu.cn

**Abstract**

The seeding effect of upper clouds on lower clouds affects the evolution of clouds, especially the seeding from upper ice clouds on lower stratiform clouds or convective clouds, which can stimulate the precipitation of lower clouds and even produce extreme precipitation. When seeders of the seeding cloud enter feeding cloud, the interaction between cloud particles results in the change of macro and micro parameters of feeding cloud. Based on the observation data of the ground-based Ka-band millimeter-wave cloud radar (MMCR) and microwave radiometer (MWR) in spring and autumn from 2021 to 2022, the seeder-feeder phenomenon among double-layer clouds in Xi'an, China, is studied. The study on 11 cases of seeder-feeder processes shows that the processes can be divided into three types by defining the height difference (HD) between the seeding cloud base and the feeding cloud top, and the effective seeding depth (ESD). Through the analysis on the reflectivity factor ($Z$) and the radial velocity ($V_r$) of cloud particles detected by the MMCR and on the retrieved cloud dynamics parameters (vertical velocity of airflow ($V_a$) and terminal velocity of cloud particles ($V_f$)), it is shown that the reflectivity factor and particles terminal velocity in the cloud are significantly enhanced during the seeder-feeder period for the three types of processes. But the enhancement magnitudes of the three seeder-feeder processes are different. The results also show that the impact of seeding on the feeding cloud is limited. The lower the height and thinner the thickness of the HD, the lower the height and thicker the thickness of ESD. On the contrary, the higher the height and the thicker the thickness of the HD, the higher the height and the thinner the thickness of the ESD.

**Keywords:** Macro parameters of cloud, natural seeder-feeder process, Ka-band millimeter-wave cloud radar, remote sensing and sensors

## 1. Introduction

Natural ice crystals in upper clouds can be the source of seeders for lower clouds (Korolev et al., 1999; Heymsfield et al., 2013; Myagkov et al., 2016; Cheng et al., 2020; Wang et al., 2023). This seeder-feeder process is able to promote the development of the lower clouds even to stimulate extreme precipitation (Choularton et al.,

1986; Locatelliet al., 1983; Robichaud al., 1988; Fernández-González et al., 2015; Ramelli et al., 2021). The seeder-feeder process is a phenomenon that ice crystals as seeder, from the upper clouds fall into the lower clouds or the lower-lying part of the same clouds, which is either liquid, ice or mixed phase (Hall et al., 1976; Korolev et al., 2003; Hong et al., 2005; Geerts et al., 2015; Lowenthal et al., 2018). When ice crystals meet lower cloud droplets with ice phase or supercool water state, they grow by riming or vapour deposition via the Wegener-Bergeron-Findeisen process vapour (Bergeron 1935; He et al., 2022). Therefore, it is important to understand the seeder-feeder process, which can be helpful to improve the representation of cloud processes in weather and climate models, and weather forecasts of precipitation, and ultimately to reduce uncertainty in climate simulations (Hong et al., 2012; Proske et al., 2021). The seeder-feeder process has been studied through observations and simulations in operations of the artificial precipitation enhancement, and it was found that the distinct changes in both cloud and precipitation properties (French et al., 2018; Ramelli et al., 2021; Dong et al., 2021).

Historically, Braham (1967) noted the natural phenomenon of ice crystals from the upper cirrus clouds acting as seeders for ice formation in warmer clouds below. It was found that not only cirrus but also altocumulus and altostratus, which contain ice crystals, may act as the seeding clouds. In the 1980s in China, Hong et al., (2012) established a cloud model that simulated the formation of stratiform clouds. In the model, the seeder-feeder process was emphasized. Subsequently, this cloud seeding process through sedimenting ice crystals has been observed in a multitude of remote sensing and aircraft campaigns. Seifert et al., (2014) and He et al., (2022) estimated the occurrence frequency of the natural cloud seeding through analyzing their lidar datasets. Furthermore, a regional occurrence frequency of seeder-feeder in the Arctic was estimated by Vassel et al. (2019). They pointed out that the seeder-feeder process happened usually within multi-layer clouds, which was observed by radiosonde and radar in Svalbard. By using the DARDAR satellite products and sublimation calculations, Proske et al., (2021) also studied the occurrence frequency of cloud seeding in Switzerland and found the high occurrence frequency of seeding situations with the survival of the ice crystals. The microphysical parameters of the seeder-feeder process appeared within mixed-phase clouds have been investigated by using the ground-based remote sensing instruments (Ramelli et al., 2021). However, there is still the lack of the specific characteristics, such as the height difference between the seeding cloud base and the feeding cloud top (HD) and the effective seeding depth (ESD), to represent the feature of the seeder-feeder process. In the meantime, the characteristic of air vertical motion, particle terminal velocity inside cloud during seeder-feeder process are still poorly understood.

The nature seeder-feeder process within clouds is not well documented in the articles (Hill et al., 2007; Purdy et al., 2005; He et al., 2022). The main reason is that the effects of the seeder-feeder process are not easy to be

measured, because several cloud layers need to be able to be monitored simultaneously with high vertical and
temporal resolution. The active instrument of the Ka-band millimeter-wave cloud radar (MMCR), a useful tool for
cloud observations, can detect multiple cloud layers directly, which allows measure the seeder-feeder process
(Ramelli et al., 2021; Proske et al., 2021). The Doppler spectra recorded by the MMCR can be used to retrieve the
vertical airflow velocity and the terminal velocity of cloud particles and to obtain information of particle types
(Luke et al., 2013; Shupe et al., 2008; Kollias et al., 2002 and 2011). However, such direct observations of ice
crystal formation and evolution in the seeder-feeder process are limited (French et al., 2018).
In this study, the seeder-feeder process happened between bilayer stratiform cloud in Xi'an are investigated by
using the observation data from the MMCR t, microwave radiometer (MWR) and radiosonde from January 2021 to
December 2022. In this paper, following the above review of study status on the seeder-feeder process, the used
instruments and methods associated with datasets are introduced simply, then through a case analysis of seeder-
feeder process measured by the MMCR to expose the evolution mechanism of seeding and feeding clouds. The
main results and conclusions will be represented by statistics with two years data.
**2. Instruments and methods**
The instruments used in this study are the MMCR, MWR, radiosonde and Raindrop Spectrometer (RDS). The
MMCR is the Doppler vertical pointing cloud profile radar with solid-state transmitter. The main parameters of the
MMCR are shown in Table 1. The MMCR can observe reflectivity factor ($Z$), radial velocity ($V_r$), spectral width
($\sigma_v$) and Doppler spectrum. These parameters can be used to retrieve cloud dynamic parameters, such as cloud
particle terminal velocity and vertical airflow velocity inside the cloud (Liu et al., 2019; Yuan et al., 2022; Di et al.,
2022). Because of the advantages of solid-state transmitter, the MMCR is small in size, long in life and good in
reliability, so it provides reliable observation data for this study. Due to the MMCR has certain penetrating ability
to cloud, it can detect the structure variation of multi-layer cloud system, so the phenomenon of seeder-feeder
between multi-layer cloud system can be measured, which is an important reason for us to choose this instrument in
this study.
The MWR includes 21 water vapour channels (distributed in the K band, that is, 22 – 31 GHz), 14 air
temperature channels (distributed in the V band, that is, 51~59 GHz), and 1 infrared channel. The observation data
of the MWR can be used to retrieve the profiles of atmospheric temperature (/K) and relative humidity (/%),
integrated water vapor content ($V_{int}$ /mm) and integrated liquid water content ($L_{qint}$ /mm). Below the height of 2 km,
the root mean square error (RMSE) of temperature measurement is less than 1 K, the RMSE of temperature
measurement is less than 1.8 K above 2 km height. The RMSE of relative humidity is less than 15 %, and the
RMSE of $V_{int}$ vapouris less than 4 mm. Table 2 lists the major technical parameters of the MWR.
Table 1 Major technical parameters of the MMCR

| Order | Items | Technical specifications |
|---|---|---|
| 1 | Radar system | Coherent, pulsed Doppler; solid-state transmitter; and pulse compression |
| 2 | Radar frequency | 35 GHz ± 200 MHz (Ka-band) |
| 3 | Antenna aperture | ≥1.6 m |
| 4 | Horizontal and vertical beam width | 0.4°and 0.4° beam width |
| 5 | Antenna gain | 53 dB |
| 6 | Pulse repeat frequency | 8000 Hz |
| 7 | Peak power | ≥20 W |
| 8 | Detecting parameters | $Z$, $V_r$, $\sigma_v$, and Doppler spectrum |
|  | Detection capability | ≤ −35 dBz at 5 km |
| 9 | Range of detection | Height: 0.15 − 15 km |
|  |  | Reflectivity factor: −45 − + 30 dBz |
|  |  | Radial velocity: −15 − 15 ms$^{-1}$ |
|  |  | Spectral width: 0–15 ms$^{-1}$ |
| 10 | Spatial and temporal resolutions | Temporal resolution: 5 s |
|  |  | Height resolution: 30 m |
| 11 | Pulse width | 1 μs; 5 μs; and 20 μs |

Table 2 Major technical parameters of the MWR

| Order | Items | Technical specifications |
|---|---|---|
| 1 | Range of detection | 0 − 10 km |
| 2 | Height resolution | ≤ 25 m (0~ − 500 m) |
|  |  | ≤ 50 m (500 − 2000 m) |
|  |  | ≤ 250 m (2 − 10 km) |
| 3 | Layering | ≥83 layers |
| 4 | Channel frequency | K-band: 22 − 31 GHz |
|  |  | V-band: 51 − 59 GHz |
| 5 | Number of channels | number of water vapour channel: 12 |
|  |  | number of temperature channel: 14 |
|  |  | number of infrared channel: 1 |
| 6 | Absolute brightness temperature accuracy | ≤1.0 K |
| 7 | RMSE of temperature profile | ≤ 1.8 K, Range >2 km |
|  |  | ≤ 1.0 K , Range ≤2 km |
| 8 | RMSE of relative humidity | ≤15% |
| 9 | RMSE of V$_{int}$ vapour | ≤4 mm |

The above instruments are placed at the Jinghe National Meteorological Station (108°58'E, 34°26'N, in Fig. 1) in Xi'an, Shaanxi Province, China, which is located near the north bank of the Wei River in Guanzhong Basin (between 107°40'–109°49'E, 33°42'–34°45'N, about 400 meters above sea level) in the middle of the Yellow River Basin. The Qinling Mountains on the south side of the Wei River often block the cold air southward in winter and spring, and produce stable stratiform clouds in the Guanzhong Plain, which provides a natural experimental site to study the seeder-feeder phenomenon in bilayer stratiform clouds. The distance between the MMCR and MWR is less than 5 m, and the distance between other instruments is less than 50 m. The Jinghe National Meteorological Station is also the national Meteorological sounding Station. Sounding balloons are launched every day at 07:15 and 19:15 BJT (Beijing Time) to detect atmospheric temperature, relative humidity, wind speed and wind direction from the ground to an altitude of 30 km in the station (Görsdorf et al., 2015; Vassel et al. 2019;Yuan et al. 2022).

The collaborative detection of the above instruments provides effective data support for searching  the seeder-
feeder process in clouds in this study.

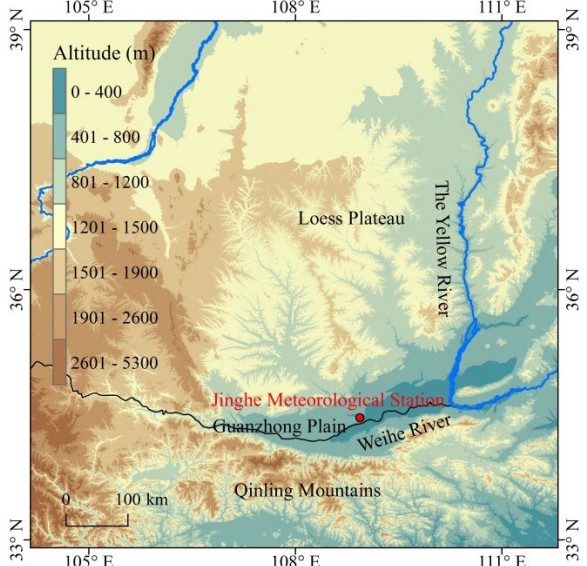

Figure 1. Geographical coverage around observation site (104°10'~111°40'E, 33°-39°N). The red dot indicates the location of the
Jinghe National Meteorological Station in Xi'an (107°40'-109°49'E, 33°42'-34°45'N).
As the MMCR adopts vertical upward mode observation, its echo signal will not be affected by ground clutter,
which reduces errors of the terrain clutter in observation data. However, due to the influence of aerosol and insects
in lower atmosphere, there will be non-cloud signals in the bottom echo signal of the MMCR. The non-cloud echo
signals in the low-level atmosphere have the characteristics of the small reflectivity factor, low radial velocity, and
large spectral width. To further eliminate interfering wave information, we obtain the data quality control threshold
by counting the characteristic changes in planktonic echoes in the boundary layer under cloud free conditions (Yuan
et al., 2022). The Doppler spectrum from the MMCR includes the information of the cloud particle size and the air
vertical motion. The Doppler spectrum is also affected by radar beam width, wind shear and atmospheric
turbulence. Therefore, radiosonde data combined with the MMCR hardware parameters are used to correct the
broadening of the Doppler spectra, to improve the accuracy of the retrieved vertical velocity of airflow ($V_a$) and the
particles terminal velocity ($V_f$) in clouds (Doviak and Zrnic, 1993; Shupe et al., 2008; Kollias et al., 2001 and 2002;
Shupe et al., 2008 and 2004).
To calculate the vertical velocity of the airflow in the cloud more accurately, the cloud phase state need be
judged. The terminal velocity of cloud particles varies due to the influence of phase state, which in turn affects the
magnitude of vertical airflow velocity. Cloud particle phase identification adopts cluster analysis method (Shupe,
2007). The specific process takes cloud reflectivity factor, particle radial velocity and spectral width  measured by
the MMCR, and atmospheric temperature measured by MWR as input parameters for cloud phase identification.
Through unsupervised learning, cloud particles of different phase states in the cloud are identified, such as warm
clouds, mixed phase (ice dominated phase or water dominated phase), ice, snow, supercooled water, drizzle, rain
and graupel particles.
In the identified ice particle region and mixed phase region of stratiform cloud, the turbulence inside the cloud is
very small and can be ignored, and the left endpoint of the Doppler spectrum represents the information of the
smallest particle detected by the radar. If particle size is small enough to ignore its terminal velocity, the left
endpoint of the Doppler spectra can be used to retrieve vertical airflow velocity, that is, the small particle tracer
method (Shupe et al., 2008). In this study, the echo intensity of –21 dBZ is used as the threshold for radar detection
volume containing small particles. When the echo intensity is less than –21 dBZ, it can be considered that the
particle size is small enough to be used as the tracer particle to retrieve vertical airflow velocity. Meanwhile, if the
spectral width is less than 0.4m/s, it is considered that the turbulence is small and can be ignored.  In the identified
supercooled water region, the peak position of the liquid cloud particle is used to obtain the vertical velocity of
airflow (Wei et al., 2019). When it drizzles, the Doppler spectra of the MMCR usually show the bimodal
distribution, and the vertical velocity of the airflow in the cloud can be obtained by the bimodal position of the
liquid cloud particles (Wei et al., 2019; Luke et al., 2010 and 2013). The radial velocity is the combination of the
particle terminal velocity and the air vertical motion. Therefore, the cloud particle terminal velocity can be obtained
by subtracting the vertical airflow velocity from the radial velocity. The terminal velocity of cloud particles and the
vertical velocity of airflow are important parameters in analyzing the seeder-feeder process. Based on the
observation data of the MMCR from 2021 to 2022 (a total of 10363 hours), the seeder-feeder process of bilayer
cloud system (ice phase in upper cloud and mixed phase cloud in lower cloud) is analyzed below.
**3. Parameter Definition and Case Analysis**
To analyze conveniently and clearly the seeder-feeder process of bilayer clouds in Xi 'an, and find how the upper
seeding clouds to seed the lower feeding clouds in this study. We have chosen observation data from the MMCR
and MWR in winter and spring, as most of the clouds in these seasons are stable stratiform clouds. The first step is
to define the relevant parameters to describe the characteristics of the bilayer clouds, such as the top height of
seeding cloud (THSC) and the base height of seeding cloud (BHSC), the top height of feeding cloud (THFC). The
height difference (HD) between the BHSC and THFC is also defined. The HD can display directly one of the
geometric features of the bilayer clouds. The heights of cloud top and base are determined from radar echo signals.
Before determining the two heights, the clutter mixed in signals observed by the MMCR were filtered out. The
sensitivity threshold of the radar used in this study is -40dBZ, which is sufficient for accurately observing the
positions of cloud base and cloud top (Yuan, et al. 2022).
A period of stable time from the moment when the stable double-layer cloud appear until the start of seeding is
denoted as t1, the moment when the seeding cloud began to seed is marked as T0, the length of time period of the
seeding is denoted as t2, and the period after the end of the seeding but the lower part of the feeding cloud still
showed development changes in the reflectivity factor is labeled as t3 (which is called the duration of the seeding
effect). Usually, the cloud base or cloud top is not flat enough. However, as our study focuses on stable stratiform
clouds, the cloud top and cloud base observed in these cases are relatively flat. The THSC is the average height of
seed clouds top during the observation period, the BHSC is the average height of the seeder clouds base during the
t1 period, and the THFC is the average height of the feeder clouds top during the t1 period.
Fig. 2 shows the cloud reflectivity factor and radial velocity detected by the MMCR from 23:00 BJT on 05[th]
February 2022 to 04:00 BJT on 06[th] February 2022. The reflectivity factor (in Fig. 2a) clearly shows the seeder-
feeder process. The period from 00:40 to 02:20 BJT, cloud particles of the upper cloud fell into the top of the lower
cloud. This is confirmed by the cloud particle radial velocity (in Fig. 2b), which shows that the cloud particles
during the period were all sinking, and the sinking velocity was about $-1 \text{ms}^{-1}$. The above defined parameters have
been marked in Fig. 2a. It also shows that the bilayer cloud are stable during this period, with THSC at 8 km,
BHSC at 5.5 km, THFC at 4.2 km, HD at 0.85 km. The seeding process lasted for about 98.2 minutes (t2), and
feeding cloud development duration reached more than 2 hours and 30 minutes. Before seeder-feeder process of the
bilayer clouds, only 40 minutes (t1) was considered as the earlier state of beclouds, and table 3 recorded the
detailed information of these parameters.

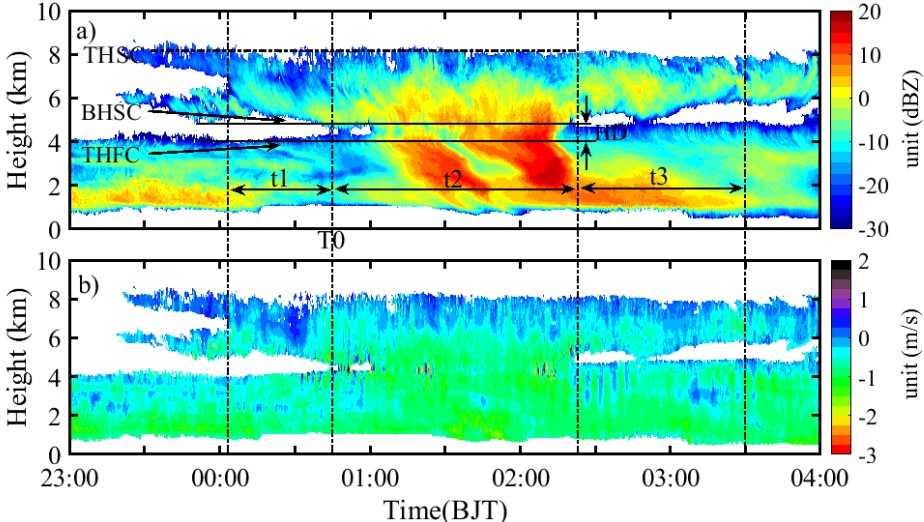


Figure 2. The variations with time for both profiles of cloud reflectivity factor (a) and radial velocity (b) detected by the MMCR from
23:00 BJT on 05[th] February 2022 to 04:00 BJT on 06[th] February 2022 (positive value in color bar represents ascending motion and
negative value represents sinking motion). The THSC and BHSC are the cloud top height and cloud base height of the seeding cloud,
the THFC is the cloud top height of the feeding cloud, and the HD is the height difference between the BHSC and THFC. The T0 is
the moment when the seeding cloud began to seed, the t1 is the stable time period before the seeding cloud begins to seed, the t2 is the
length of time from beginning to end of the seeding, and the t3 is the period after the end of the seeding but the reflectivity factor in the
feeding cloud still development.
Table 3 Values of the defined parameters for the seeder-feeder process shown by Fig. 2

| Parameters | THSC /(km) | BHSC /(km) | THFC /(km) | HD /(km) | t1 /(min) | t2 /(min) | t3 /(min) |
|---|---|---|---|---|---|---|---|
| Values | 8.2 | 5.1 | 4.3 | 0.85 | 40.2 | 98.2 | 44 |

In order to reveal the variation characteristics of the cloud system during this seeder-feeder process, the spectral
width of cloud particles, the vertical airflow velocity, and the terminal velocity of cloud particles are firstly
calculated from the signals of the  Doppler spectra detected by the MMCR (in Fig. 3, the  positive indicates being
away from radar, and negative indicates pointing to the ground). Fig. 3a shows that the spectral width was small,
indicating that the cloud particle radial velocity detected by the MMCR was relatively stable, which also indicates
that the airflow inside bilayer clouds was stable. The maximum value of spectral width  was approximately0.6 ms$^{-1}$,
which was mainly located at the top of the seeding and feeding clouds (especially at the beginning of the feeding
cloud), and the lower part between the seeding and the feeding clouds during the seeding period (that is, the top of
the feeding clouds). In addition, the feeding clouds showed changes in the t3 period after seeding, that is, the
feeding cloud top height rose slightly (in Fig. 2a), and the spectral width  increased at the cloud top region, which
indicated that the radial velocity at this region changed greatly during the t3 period. This is probably because of
latent heating release by the phase transition in the seeding clouds during the seeder-feeder process, which will be
feedback the dynamic process, then increases the vertical velocity of the airflow inside the cloud. This position in
Fig. 3b indeed indicates that the vertical velocity of the airflow was relatively large (0.5~2 ms$^{-1}$). Fig. 3b shows that
weak upward movement (0.5~2 ms$^{-1}$) prevails in the seeding and the feeding clouds, which was consistent with the
dynamic structure characteristics of stable stratiform clouds (Hou et al., 2010; Wang et al., 2022) in winter and
spring in Xi'an. The maximum vertical velocity of airflow was located at the junction of upper and lower clouds,
the top and base of seeding clouds and the top of feeding clouds in t3 period. During the seeding period, there were
the large air upward movement (up to 1.5 ms$^{-1}$) in the middle and lower regions of the feeding clouds. There was
rarely a large-scale and prolonged air upward and downward movement in the seeding and feeding clouds, but
alternating upward and downward movement occurred.
Fig. 3c clearly shows the terminal velocity of cloud particles, and it was in the range of -1~ -4 ms$^{-1}$ during the
seeding process (the negative represents the downward movement in Fig.3c ), but most of them were less than 2.5
m/s. During this seeding process, 00:45~01:50 BJT and 02:00~02:20 BJT were two significant seeding periods, and
the maximum terminal velocity of cloud particles was about 4 ms$^{-1}$ in last period, which indicates the large cloud
particles size. According to the cloud phase in Fig. 4, the particles were snowflakes in the cloud seeding and
feeding areas. The particle size is related to the shape of snowflakes and the terminal velocity, so it difficult to
accurately quantify particle size. The relations of snow particles and diameter were studied in the ref. (Tao et al.
2020). According to the speculation, the diameter of the snow particles in the clouds was distributed between 1 mm
and 6 mm, and most of them were below 3 mm. In the areas unaffected by the seeding (except for the bottom area

of the lower clouds during the period of 23:00~24:10 BJT), the particle terminal velocity was small, less than 1.5 ms$^{-1}$. These all indicates that the seeding had the significant enhancing effect on particle diameterof the feeding clouds.

According to Table 3, the HD between seeding and feeding clouds was 0.85 km. If the sinking speed of cloud particles was at –1 ms$^{-1}$ ( in Fig. 2b), it take about 14 minutes for cloud particles to fall from the seeding cloud base to the feeding cloud top. In addition, Fig. 2 and Fig.3 show that the seeding end at 02:20 BJT, but Fig. 3c still shows that after this time, cloud particles still sink (, sinking velocity was approximately –0.5 ms$^{-1}$ at 02:45 BJT) on the feeding cloud top. It is likely that the MMCR is limited in its sensitivity to detect smaller particles and cannot clearly show the reflectively factor of small particles. The above results indicate that the sinking motion region of the cloud particle  can be used to identify the seeding cloud effectively. Anyway, the above gives an important conclusion, i.e. after seeding, the feeding cloud top rose slightly, which may be the result of latent heating release. The sinking motion region of particle  can directly characterize the seeding process.

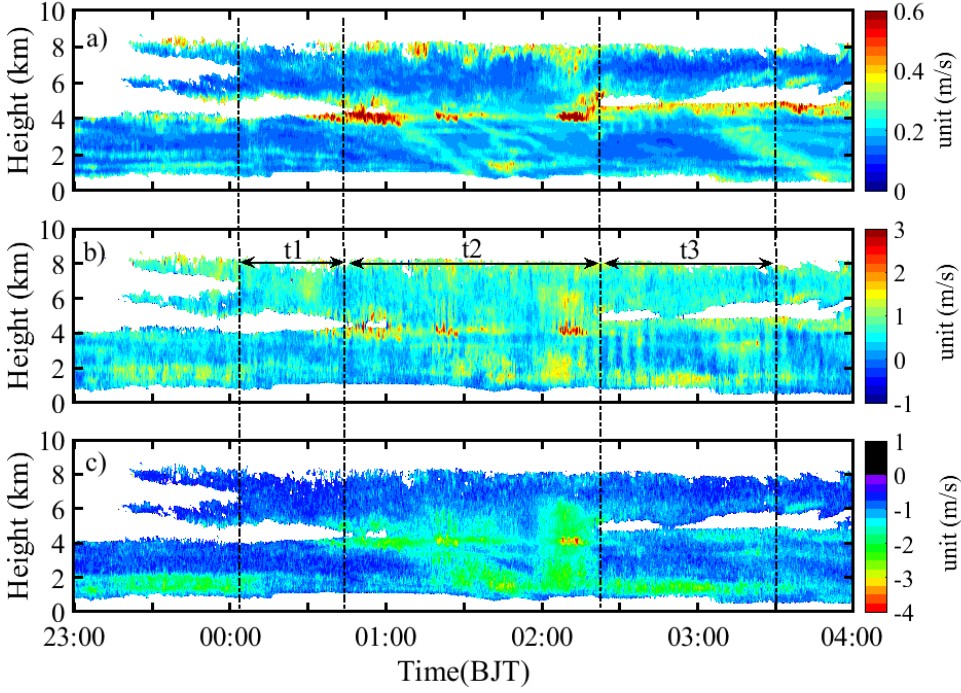

Figure 3. The  spectral width of cloud particles (a), vertical velocity of the airflow (b), and the terminal velocity of cloud particles (c) based on the retrieval from the MMCR (positive represents ascending motion and negative value represents sinking motion).

By using the observation data of the MMCR and MWR, the cloud phase state, water vapour structure, and the total amount of liquid water and water vapour in the column can be retrieved. Fig. 4a shows that seeding clouds consisted of ice and snow, and seeding was caused by sinking ice particles. Before seeding, the particles in the feeding cloud were basically in mixed phase, and there was a thin layer of supercooled water in the middle and upper part of the cloud, and snow particles appeared at the base of the cloud for a short time after 00:10 BJT. Before seeding, the larger downward radial velocity (in Fig. 2b) was detected in the lower part of the seeding cloud, which indicates that the cloud process has transformed from ice to snow with large particle diameter. Snowflakes,

as seeders, fall into the mixed phase cloud containing supercooled water, so that the Wegener-Bergeron-Findeisen effect or accretion occurred. That process caused the supercooled water in the mixed phase cloud to rapidly transform into ice. Because it takes time for particles to fall, so the seeding continued to the middle and lower parts of the feeding clouds, and snow keep for a long time (maintaining the entire t3 period); In the top region of the unaffected feeding cloud, the cloud phase remained supercooled water, which was consistent with Shupe's (2007) observation results . The temperature of the supercooled water layer was approximately –20°C, while that of the seeding cloud top was close to –40°C. From Figure 4, it can be seen that the instantaneous water vapour flux of the seeding clouds was smaller than that of the feeding clouds, vapour and the bottom layer of the feeding clouds had the instantaneous water vapour flux greater than 20 $\mathrm{gm^{-2}s^{-1}}$, indicating that the lower layer of the atmosphere had high humidity during the seeder-feeder process in the bilayer stratiform clouds.

The temporal variation of column water vapour and column liquid water given by the MWR (in Fig. 4c) showed that both rapidly increased from t1 before seeding to the beginning of seeding, and rapidly decreased after seeding. Before the second intense seeding, column water vapour and column liquid water content increased rapidly, and then decreased with the end of seeding. This process can be understood as that when the ice phase particles from the seeding cloud entered the supercooled water of the feeding cloud top, the Wegener-Bergeron-Findeisen effect or accretion was triggered, and the liquid particles were rapidly transformed into ice phase particles, which led to the reduction of liquid water content in the column. The above results illustrate that the seeders from the seeding clouds caused change of cloud phase state in the feeding clouds, thus reducing the water vapour and liquid water in the column. The rapid increase of water vapour and liquid water in the column before seeding was related to the change of atmospheric environment at that time, which still needs to be studied in detail.

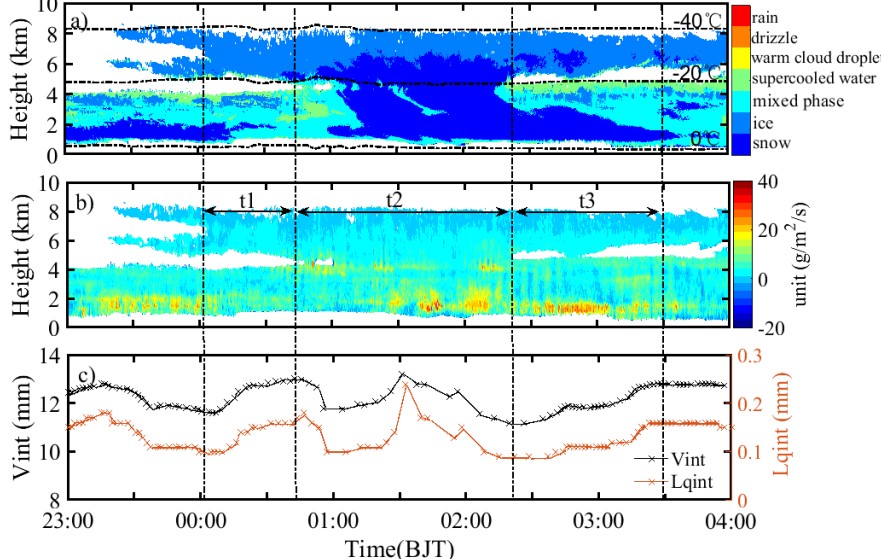

Figure 4. The variations of cloud phase (a), water vapour flux (b), integrated water vapour content ($V_{int}$, shown by the black line) and integrated liquid water content ($L_{qint}$, shown by the orange line) (c) with time from the MMCR and MWR.

According to the radar formula, the echo signal intensity is proportional to the sixth square of the cloud particle

diameter. The cloud particle with larger diameter has a larger falling velocity under the action of gravity. To reveal
the relationship between particle diameter and echo signal in the seeder-feeder process. The statistical classification
method of equal samples was adopted to find the relationship. All signal values (reflectivity factor, radial velocity,
spectral width, particle terminal velocity, and vertical airflow velocity) were reordered according to their
corresponding the reflectivity factor from small to large, and then compared in the equal sample. For example, the
first 33%, middle 33% and last 33% of the samples were arithmetically averaged to obtain the mean reflecting the
weak, moderate and strong values respectively. This has the advantage of avoiding the defect of large and small
arithmetic averages cancelling each other out. Following this principle, the reflectivity factor of t1, t2 and t3 were
arranged in ascending order, and the corresponding parameters of cloud particles were also sorted with the order of
the reflectivity factor, and then the arithmetic average was performed according to the first 33%, middle 33% and
last 33% of the samples. The average profiles representing weak echo, moderate echo and strong echo were
obtained(in Figs 5a1, a2, a3), and the corresponding average profiles of cloud particle parameter for the three
intensity echoes were also obtained, and they were the corresponding average profiles of radial velocity (in Figs
5b1, b2, b3), average profiles of spectral width (in Figs 5c1, c2, c3), average profiles of particle terminal velocity
(in Figs 5d1, d2,d3) and average profiles of vertical airflow velocity (in Figs 5e1, e2, e3).

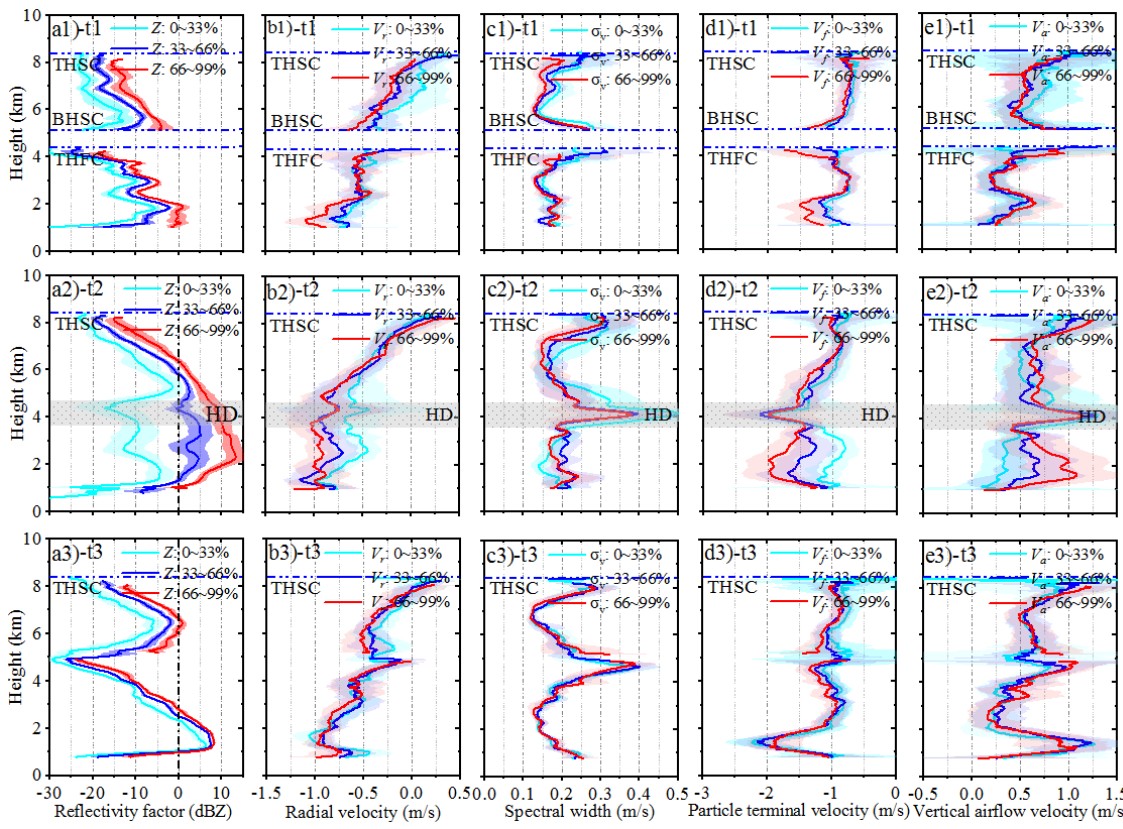

Figure 5. The mean profiles of reflectivity factor (a1, a2, a3), radial velocity (b1, b2, b3), spectral width (c1, c2, c3), particle terminal
velocity (d1, d2, d3) and vertical velocity of the airflow (e1, e2, e3) during t1 (up), t2 (middle) and t3 (bottom) periods respectively. In
the figure, the cyan line, blue line and red line represent the average of the first 33%, the middle 33% and the last 33% of the sample
respectively The solid line represents the mean, and the shaded area of the corresponding color is the variance.
The up panel of Fig. 5 shows that there were obvious differences (in Fig. 5 a1) between the weak, moderate and
strong reflectivity factor profiles of the seeding clouds and feeding clouds before the seeding (in t1 period), but in
general, the average profiles of the three kinds of echo intensity show that reflectivity factor increased with the
decrease of detection height, and the values of the profiles were relatively small (all less than 0 dBZ) and the
variance was also small. However, the profiles of cloud particle radial velocity (in Fig. 5 b1), spectral width (in Fig.
5 c1), particle terminal velocity (in Fig. 5 d1) and vertical airflow velocity (in Fig. 5 e1) corresponding to the
average profiles of the three intensity reflectivity factors basically coincided, and did not show significant changes
in these parameters caused by differences in reflectivity factors. This indicates that the cloud particle states (radial
velocity, spectral width, particle terminal velocity and vertical velocity of airflow) of the seeding and feeding
clouds in t1 period were uniformly distributed at different intensity echoes, that is, the upper and lower cloud
systems were stable before seeding, and the cloud particle diameter was mainly small.
The middle panel of Fig. 5 represents the average profiles of each parameter in the seeding period (t2). Fig. 5 a2
shows that the difference between the average profiles of the reflectivity factor for the three kinds of echo intensity
was greater than that before seeding. In particular, the profiles of the moderate and strong reflectivity factor in the
figure increased significantly, reaching the maximum of 15 dBZ, which hinted that the diameter distribution of
cloud particles varied significantly during seeding in the bilayer cloud. Compared with before seeding, the
reflectivity factor of the lower part of the seeding cloud (5.4 km ~ 6.2 km) increased significantly, and they
increased from the initial values of –20dB ~ –5dBZ to the values of –10 dBZ ~ 10 dBZ. The absolute value of
radial velocity (in Fig. 5 b2) and the terminal velocity of cloud particles (in Fig. 5 d2) increased significantly, and
they reached ~ 0.8m/s and ~ 1.3m/s before falling into the feeding clouds.  These changes of seeding clouds
particles produced the seeding effect. The spectral width of feeding clouds p (Fig. 5 c2) corresponding to the three
intensity reflectivity factors didn't coincide, which changed significantly with particle distribution or types. For the
strong reflectivity factor profile, from the top of the seeding clouds to the lower part of the feeding clouds at the
height of 2 km, reflectivity factor increased rapidly with the decrease of the detection height. The corresponding
radial velocity and particle terminal velocity increased (i.e. the descending velocity increases), reaching the
maximum of 0.9 m/s and 2 m/s, respectively and the vertical airflow velocity also increased, reaching the
maximum of 1m/s. Those indicate that the large particles in the seeding clouds had a great effect on the feeding
clouds. For the weak reflectivity factor profile of bilayer cloud, the average reflectivity factor changes little
compared with that before seeding, indicating that the seeding effect of small cloud droplets corresponding to such
weak echoes was small. Fig. 5 also shows that during the seeding period, the reflectivity factor of the middle and
upper part of the feeding clouds increased significantly after the seeders were injected into the feeding clouds,
especially in the case of strong and moderate intensity, indicating that the middle and upper part of the feeding
clouds particle diameter became significantly larger, which clearly expressed the seeding effect.
With the end of seeding (bottom panel in Fig. 5), the reflectivity factor of the upper and middle part of the
seeding clouds decreased significantly to less than ~ 0 dBZ. The reflectivity factor of the lower part of the feeding
clouds increased, reaching the maximum of ~8 dBZ at the height of 1.5 km~2 km, which reveals that the seeding
effect developed to the lower part of the feeding clouds. In general, the distributions of strong, moderate, and weak
reflectivity factor profiles in feeding clouds were concentrated after seeding, informing that cloud particle diameter
became more uniform, which was obviously different from that before and during seeding. Therefore, the profiles
of cloud particle radial velocity, spectral width, particle terminal velocity and vertical airflow velocity
corresponding to the strong, moderate, and weak reflectivity factor basically coincided. Since reflectivity factor,
radial velocity, and particle terminal velocity reflect particle diameter, and spectral width reflects particle diameter
distribution and particle kinds. After the end of seeding,the cloud particle diameter distribution and particle
velocity of the bilayer cloud during t3 period may reached a relatively balanced and stable state through the
complex microphysical and dynamic interactions during t2 period. However, the reflectivity factor of the feeding
clouds during t3 period reached the maximum (–8 dBZ) in the lower layer (1 km ~2 km), and the corresponding
radial velocity and particle terminal velocity of cloud particles also reached the maximum (1 m/s and ~2 m/s,
respectively), indicating that seeding effect continued at the lower part of the feeding clouds although seeding has
ended at the top of the feeding clouds. The key takeaway from Fig. 5 is that the reflectivity factor (related to cloud
particle diameter) and the descending velocity of cloud particles increased within a certain depth of the feeding
cloud during the seeding period. After the end of seeding, there was a seeding continuation period in the middle and
lower part of the feeding clouds.
To understand the effect of seeding clouds on feeding cloud, the correlation coefficient between cloud particle
terminal velocity and reflectivity factor was calculated statistically. Firstly, the correlation coefficient between the
terminal velocity of cloud particles during the t2 and t3 periods and the corresponding reflectivity factor (called the
autocorrelation coefficient, because the terminal velocity of cloud particles has a certain relationship with the size
of cloud particles, while the reflectivity factor is proportional to the $6^{th}$ power of the particle diameter) was
calculated. Therefore, the cloud particle terminal velocity is not independent of the reflectivity factor. The obtained
autocorrelation coefficient profile during the seeding period (t2) is shown in Figure 6a, which indicates that as the
detection height decreased from the middle of the seeding clouds (6 km) to the middle and lower part of the feeding
clouds (2.5 km), the autocorrelation coefficient increased from 0 to 0.8, that is, the positive correlation between the
cloud particle terminal velocity and the reflectivity factor increased continuously. The reflectivity factor also
increased with the decrease of altitude (from –5 dBZ to 5 dBZ), illustrating the reflectivity factor and terminal
velocity of cloud particles increased with the decrease of height, which may be inferred that the diameter of cloud
particles also increased with the decrease of height. As their correlation coefficient increased, which indicated that
the particle diameter was constantly increasing because of seeding. When the correlation coefficient started to

decrease, it indicated that the particle diameter also started to decrease, and the influence of the seeding began to weaken or transfer to other positions. Therefore, the position with the highest correlation coefficient is defined as the final position in the feeding cloud affected by seeding. So, the effective seeding depth (ESD) is defined as the height difference between the top height of the feeding clouds and from the height down to the height of the maximum correlation coefficient, which represents the influence of the seeders on the feeding clouds in t2 period. In this case, the ESD was about 1.6 km. In the ESD region, the reflectivity factor increased with the decrease of the detection height, so the cloud particle diameter also increased rapidly with the decrease of the height, the correlation in the middle and lower part of the ESD region  in where the seeding effect was the most intense. In the upper part of the ESD region (i.e. the top of the feeding clouds), the reflectivity factor was slightly smaller (less than 3 dBZ) and the correlation coefficient was also smaller (less than 0.2), indicating that the upper cloud particle diameter of the feeding clouds was small, and the correlation between the terminal velocity of the cloud particle and the reflectivity factor was poor, because the seeders has just entered the top of the feeding clouds and the seeding effect has just begun.

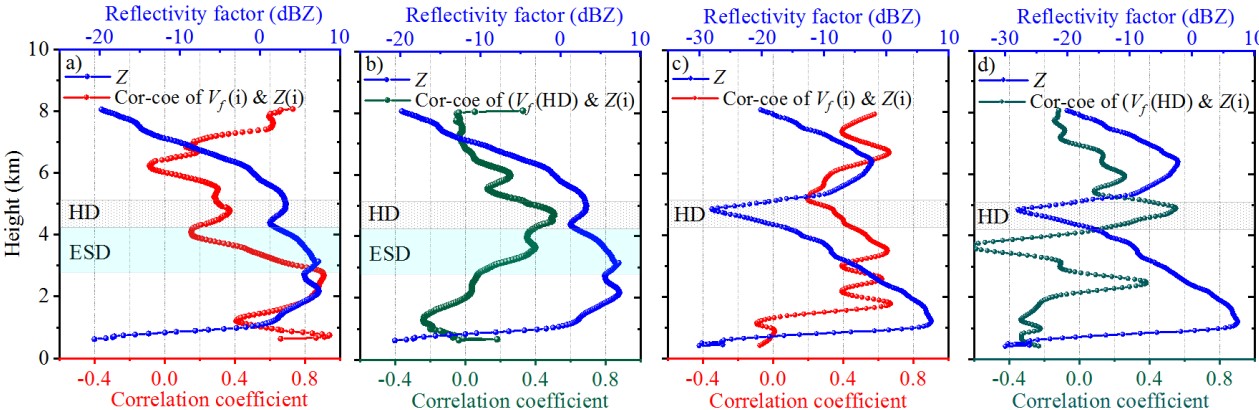

Figure 6. The autocorrelation coefficient profile (a) between cloud particle terminal velocity  and reflectivity factor at each range gate from top to bottom in the bilayer cloud in the t2 period, the non-autocorrelation coefficient profile (b) between the average descent terminal velocity in the HD region and reflectivity factor at each range gate in the t2 period, The autocorrelation coefficient profile (c) and the correlation coefficient profile (d) in the t3 period.

If the region between the upper and lower clouds, i.e. HD region, is regarded as a whole layer, the correlation coefficient between the average terminal velocity of cloud particles in this layer and the reflectivity factors of the range gate in the seeding period (t2) (called non-autocorrelation coefficient, because the terminal velocity of cloud particles and the reflectivity factor in the non-HD region are relatively independent at this time) was calculated, and the non-autocorrelation coefficient profile in Fig. 6b was obtained. It shows that above the height of the HD region, the positive correlation between the average terminal velocity of cloud particles and the reflectivity factor of each layer of the seeding cloud increased as the height decreased, indicating that the terminal velocity of cloud particles in the HD region was mainly affected by the reflectivity factor of the lower layer of the seeding cloud. The larger the reflectivity factor of the lower layer of the seeding cloud was, the larger the velocity of cloud particles in the

HD region, which conforms to the physical principle. As the height decreased to the base of the feeding cloud, the non-autocorrelation coefficient decreased from 0.4 to –0.2, indicating that the average terminal velocity of cloud particles in the HD region was only positively correlated with the reflectivity factor near the top of the feeding cloud, that is, cloud particles in the HD region only affect the clouds near the top of the feeding cloud, but had little effect on the lower part of the feeding cloud. This shows that the reflectivity factor in the middle and lower part of the feeding cloud had little correlation with the terminal velocity of cloud particles in the HD region.

The autocorrelation coefficient profile in the t3 period is shown in Figure 6c, which shows that as the height decreased from the middle of the HD region (~5 km) to the upper part of the feeding clouds (~3.1 km), the autocorrelation coefficient increased from 0.2 to 0.5, and the increase in reflectivity factor and correlation coefficient was smaller than that of t2 period at the same detection height, which indicates that the particle diameter continued to increase as the height decreases, while the impact of seeding on feeding clouds was limited due to the insufficient seeding. The reflectivity factor reached its maximum within the detection height range of 1 ~2 km, but the correlation coefficient did not increase synchronously, but oscillated and decreased, indicating that the increase in reflectivity factor was not only caused by the increase of particle diameter, but also by the increase of particle number. From Figure 6d, there was no clear correlation relationship between the non-autocorrelation coefficients and the reflectivity factor in the t3 period.

In generally, the effect of seeding clouds on feeding clouds was mainly manifested in the middle and upper part of feeding clouds, that is, the seeding effect activins in the effective seeding depth. During the seeding period, the cloud particle diameter was small (low reflectivity factor) from the top of feeding clouds upward to the 1 km height. From top to bottom in the ESD region, the cloud particles diameter increased (the reflectivity factor increased), indicating that seeding mainly occurred in this depth. After the end of seeding, the continuous influence of the seeding process in the feeding clouds can be understood as the delay of seeding benefits, and can also be understood as the seeding process inside the feeding clouds, that is, the seeding of the middle part of the feeding clouds to its lower part.

## 4. Statistical characteristics

To reveal the characteristics of the seeder-feeder process of bilayer cloud over the Shanxi-Guanzhong Plain, China, the observation results by the MMCR from winter to the next spring from 2021 to 2022 were analyzed, because a large range of compact and stable stratiform clouds often appear in the region during these seasons. During the observation period, the MMCR observed 11 cases of seeder-feeder processes of stratiform clouds. Table 4 lists the time of seeder-feeder processes, THSC, BHSC, THFC, HD, t1, t2, t3, the phase of feeding cloud base and precipitation conditions on the ground. According to the precipitation records observed by the ground rain gauge, Table 4 shows that there are 6 cases with precipitation occurrences (one with snowfall) after the seeder-feeder process occurred. In 4 cases, the base height of feeding clouds dropped to about 560 m, and the radial

velocity at the cloud base was in range of $-2 \sim -3ms^{-1}$, these cloud particles were liquid, so it should be virga (drizzle that did not fall to the ground) at the base of the feeding clouds. The process on 31[th] March 2022, the reflectivity factor of the middle and lower part of the feeding clouds increased after seeding, and the cloud particles mainly moved down. However, due to the high height of the cloud base (about 3.9 km), the retrieved cloud phase showed mixed phase, and no precipitation was observed by the ground rain gauge.

Table 4 The characteristic parameters of the seeder-feeder processes of bilayer stratiform clouds from 2021 to 2022.

| Type | Time | THSC /(km) | BHSC /(km) | THFC /(km) | HD /(km) | t1 /(min) | t2 /(min) | t3 /(min) | Phase of feeding cloud base | Precipitation state |
|---|---|---|---|---|---|---|---|---|---|---|
| I | 2021-11-29 | 10.23 | 6.00 | 5.20 | 0.80 | 101.5 | 91.3 | 114.9 | rain | Yes |
| | 2022-02-06 | 8.20 | 5.10 | 4.30 | 0.80 | 40.2 | 98.2 | 44 | ice | No |
| | 2022-02-06 | 8.43 | 5.61 | 4.86 | 0.75 | 49.1 | 113.9 | 33.6 | snow | Yes |
| | 2022-04-30 | 9.21 | 5.80 | 4.84 | 0.96 | 73.6 | 65.1 | 34.1 | rain (virga) | No |
| | 2022-11-16 | 8.79 | 5.71 | 4.77 | 0.94 | 23.7 | 36.3 | 9.0 | rain (virga) | No |
| II | 2021-01-23 | 9.45 | 6.12 | 4.50 | 1.62 | 80.3 | 59.6 | 29.5 | rain (virga) | No |
| | 2021-03-10 | 11.04 | 7.21 | 6.06 | 1.15 | 67.9 | 138.0 | 45.3 | rain | Yes |
| | 2022-03-31 | 10.02 | 7.74 | 6.25 | 1.49 | 30.3 | 30.9 | 23.3 | mixed phase | No |
| | 2022-06-04 | 10.23 | 6.99 | 5.43 | 1.56 | 15. 7 | 41.7 | 13.4 | rain | Yes |
| III | 2022-04-24 | 10.62 | 9.26 | 8.15 | 1.11 | 30.0 | 103.1 | 41.8 | rain | Yes |
| | 2022-11-08 | 10.65 | 8.04 | 5.82 | 2.22 | 35.8 | 47.0 | 17.5 | rain | Yes |

Table 5  Statistical results of characteristic parameters of three types of the seeder-feeder processes.

| Type | Samples | Variable | THSC /(km) | BHSC /(km) | THFC /(km) | HD /(km) | t1 /(min) | t2 /(min) | t3 /(min) |
|---|---|---|---|---|---|---|---|---|---|
| I | 5 | Mean | 8.97 | 5.64 | 4.79 | 0.85 | 58 | 81 | 47 |
| | | RMSE | 0.51 | 0.09 | 0.08 | 0.01 | 741 | 747 | 1282 |
| II | 4 | Mean | 10.18 | 7.02 | 5.56 | 1.46 | 60 | 68 | 28 |
| | | RMSE | 0.33 | 0.34 | 0.47 | 0.03 | 452 | 1756 | 134 |
| III | 2 | Mean | 10.64 | 8.65 | 6.99 | 1.67 | 33 | 75 | 30 |
| | | RMSE | 0.00025 | 0.37 | 1.36 | 0.31 | 8 | 787 | 148 |

Based on the characteristic parameters of seeding and feeding clouds listed in Table 4, the seeder-feeder process can be generally divided into three types according to the THSC and HD. The seeding process of type I had low seeding height ((BHSC<6 km) and small HD (HD≤1km), the type II had higher seeding height (6km≤BHSC< 8km) and larger HD (HD≥1km), and the type III also had higher seeding height (BHSC≥8 km,) and larger HD (HD≥1km). Table 5 shows the characteristic parameter distributions of these three types of the seeder-feeder processes. The average thickness of HD in the type I was 0.85 km, the average length of seeding time t2 was 81 min, and the average duration of seeding effect time t3 was 47 min (the longest among three types of seeder-feeder

processes). The average HD thickness of type III was the deepest (1.67 km), and the duration of seeding time t2 and
seeding effect duration t3 were longer than those of type II.
In order to expose the internal mechanism of the seeder-feeder process, the distribution of probability density
with height (DPDH) for the reflectivity factor, radial velocity, spectral width, particle terminal velocity and vertical
velocity of air flow in these three seeder-feeder types were calculated and plotted. Figs 7a1, 7b1 and 7c1 show the
differences in the distribution of reflectivity factor with height in the three types before seeding. The differences of
HD and its height before seeding are clearly shown, and the reflectivity factor of feeding clouds before seeding was
small. Figs 7a2, 7b2 and 7c2 clearly show that the reflectivity factor of both seeding and feeding clouds increased
during the seeding period, especially the cloud base height of the feeding clouds decreased significantly, indicating
that the development of feeding clouds caused by seeding is likely to cause precipitation. After seeding, the
reflectivity factor of the seeding clouds weakened and their thickness thinned (even disappeared in the type III), but
the lower part of feeding clouds continued to develop (in Figs 7a3, 7b3, 7c3), especially in the type I. The above
shows that when the HD was small and its height was low (type I), the seeding clouds had the greatest influence on
the feeding clouds, because in this case, the distance between the seeding and the feeding clouds was short, and the
seeders were easy to affect the feeding clouds.

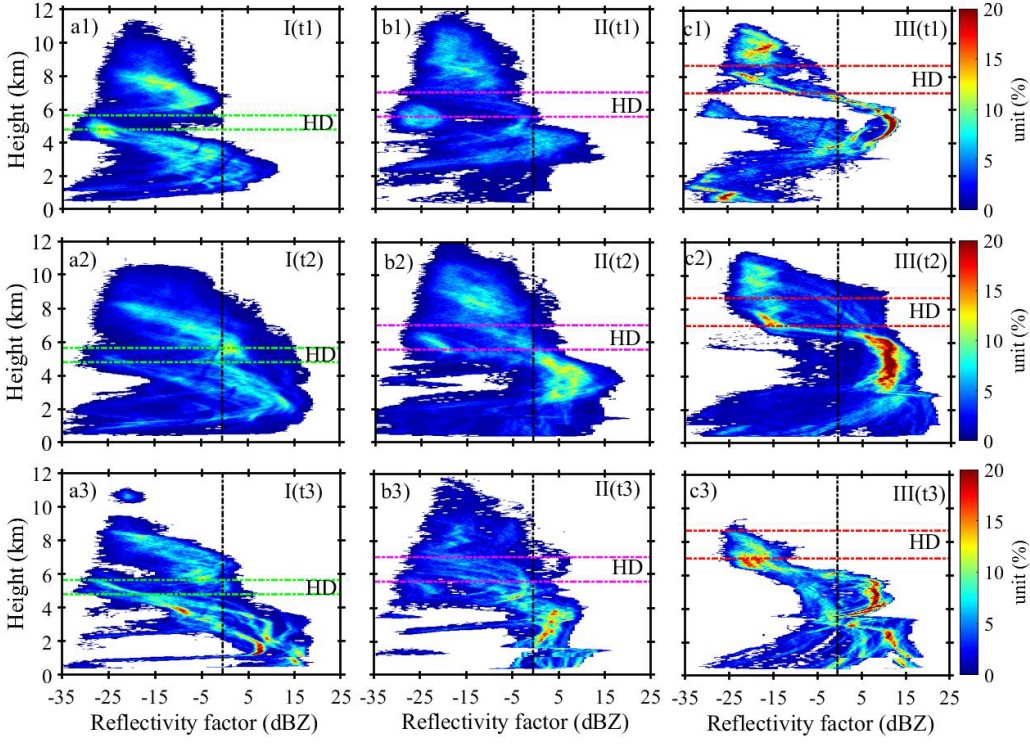


Figure 7. The distributions of probability density with height of the reflectivity factor in the three types of seeder-feeder process
before (t1), during (t2) and after (t3) seeding. The type I (5 cases) on the left column, the type II (4 cases) in the middle column, and
the type III (2 cases) in the right column. Note: the HD of type I is thin and low in height, the HD of type II is thick and slightly higher
in height, and the HD of type III is thick and the highest in height, the same below.

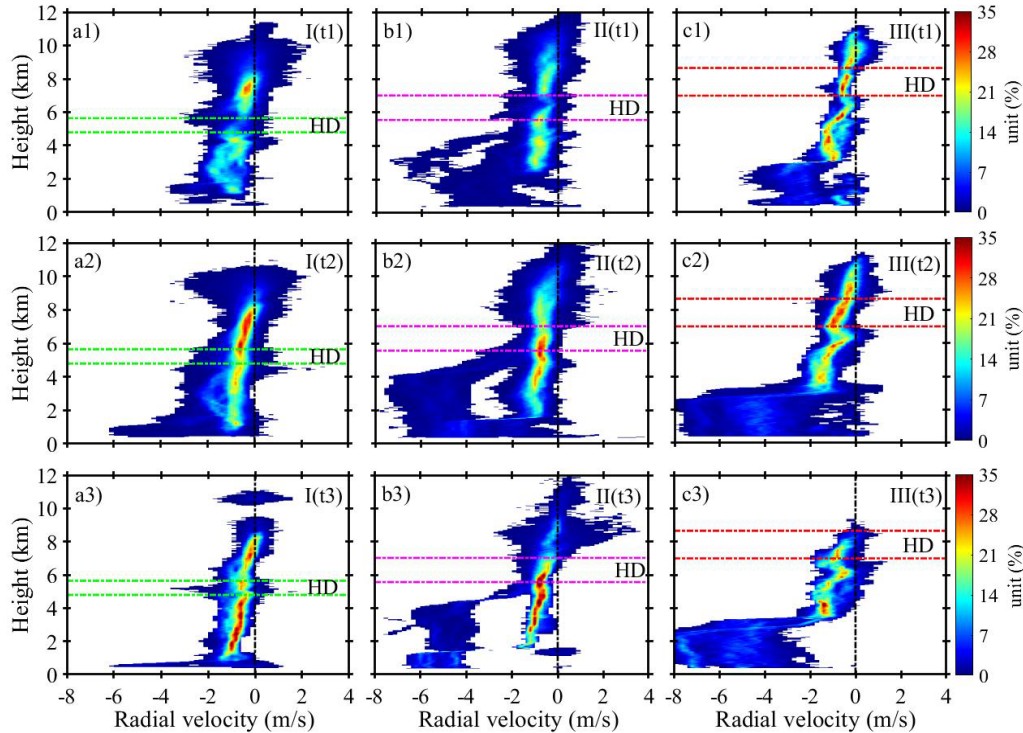


Figure 8. The distributions of probability density with height of the radial velocity in the three types of seeder-feeder process before
(t1), during (t2) and after (t3) seeding.

The cloud particle radial velocity detected by the MMCR was the actual motion velocity of cloud particles in the

clouds, which could be understood as the synthesis velocity of the vertical air flow velocity and the terminal
velocity of cloud particles. The DPDHs of radial velocity  were plotted in Fig. 8, which show a weak rising
movement in the upper part of the seeding clouds before seeding in three types, while the weak sinking movement
appeared in the lower part. In the feeding clouds, a weak subsidence existed consistently with slightly larger near
the ground. In general, the radial velocity of most cloud particles in seeding and feeding clouds kept sinking
motion, and the sinking motion increased with decreasing detection height. The radial velocity of cloud particles in
seeding and feeding clouds remained the same as before seeding. However, after seeding, the Sinking radial
velocity of cloud particles decreased (subsidence motion increased) in the both seeding and feeding clouds of the
type I, the same to the types II and III. In the meantime, the seeding clouds disappeared in the type III (consistent
with Fig. 7c3). The most important feature is that the radial velocity of cloud particles increased with the decrease
of height from before seeding to seeding process and after seeding for the three types of seeder-feeder processes.
After seeding, the  radial velocity of cloud particles in the lower part of the feeding clouds increased significantly.

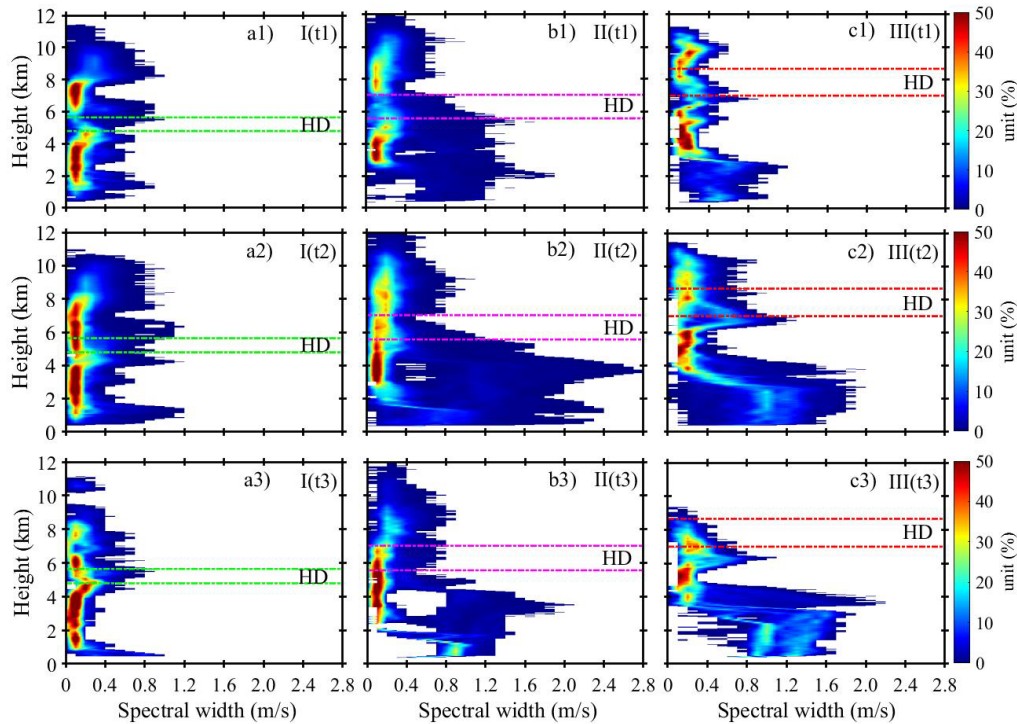

Figure 9  The distribution of probability density with height s of the spectral width in the three types of seeder-feeder process before (t1), during (t2) and after (t3) seeding.

The spectral width detected by the MMCR reflects the distribution range of cloud particle velocity. The larger value indicates the larger change in cloud particle velocity, while the smaller value indicates uniform cloud particle velocity. Fig. 9 shows the DPDHs of spectral width in the three types of seeding and feeding clouds. The figure shows that the spectral width of the most seeding and feeding clouds was less than 0.4 ms$^{-1}$, and the distribution of particle spectral width in the type I was the narrowest (most of them were less than 0.2 ms$^{-1}$). Moreover, the spectral width did not change significantly before and during seeding (in Figs 9a1 and a2). But it became significantly narrower after seeding (in Fig. 9a3), which indicates a relatively uniform of the velocity of cloud particles. That was consistent with the DPDHs of the radial velocity with height as shown in Fig. 8 a3. The spectral width of the types II and III was wider than that of the type I. The maximum of the spectral width reached more than 1.6 ms$^{-1}$, and the spectral width in the feeding clouds was wider than that in the seeding clouds, i.e. the velocity of cloud particles in the feeding clouds was greatly different. In the process of seeding, the spectral width of cloud particles for the type II and III became significantly wider (in Figs. 9b2 and c2), which is the evidence of the seeding effect resulting in the wide velocity distribution of cloud particles within the feeding clouds. After seeding, the spectral width in feeding clouds of the type II and III remained relatively wide (in Figs. 9b3 and c3). In the HD area, the spectral width was wider in the type II and III than in the type I during seeding, which may portend a wider distribution of the cloud particle diameter in the type II and III. While in the top of the feeding

clouds, there was a small spectral width for the three types of seeder-feeder process, which hints the relatively
uniform of cloud particle velocity and the narrow distribution of cloud particle diameter.

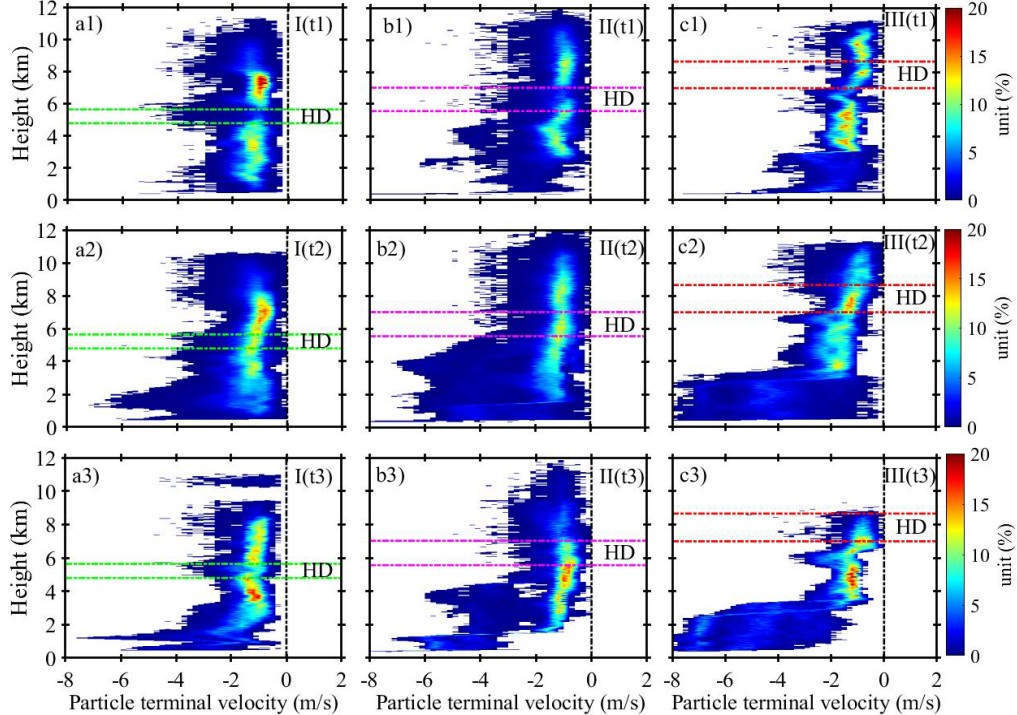


Figure 10 The distribution of probability density with height of the particle terminal velocity in the three types of seeder-feeder
process before (t1), during (t2) and after (t3) seeding.
The terminal velocity of cloud particles is the result of subtracting the vertical airflow velocity from the radial
velocity. As shown in Fig. 10, the DPDHs of the particle terminal velocity in the three types of seeding and feeding
clouds varied. In general, the particle terminal velocity of the three types was primarily distributed in the range of –
0.5 ms$^{-1}$ ~ –2 ms$^{-1}$, and the distribution of the particle terminal velocity during the seeding process (t2) and after the
seeding process (t3) was wider than that before the seeding (t1). In the seeding process, the particle terminal
velocity distribution was the widest, the maximum velocity in the type II and III were approximately –6 ms$^{-1}$ and –
8 ms$^{-1}$, respectively. The large particle terminal velocity was located at the lower part of the feeding clouds after the
seeding for the three types, which was likely to be caused by the seeding effect to increase the cloud particle
diameter in the feeding clouds. Then, under the action of gravity, the descending velocity of cloud particles
increased, and even rainfall occurred (the type III). During the seeding period of the three types (in Fig. 10a2, b2,
c2), the particle terminal velocity increased slightly with the descending height from the HD to the top of the
feeding clouds, indicating that the size of seeders in the HD region increased during the descending process and
when they entered the upper part of the feeding clouds, which reflected the seeding effect of seeders. In the middle
to lower part of the feeding clouds, the distribution of the particle terminal velocity was wide, which may be caused
by the development of the feeding clouds itself. After end of the seeding in the three types (in Figs. 10a3, b3, c3),
the particle terminal velocity increased in the middle and lower part of the feeding clouds, which could be
understood as the delay of seeding effect to the lower part of the feeding clouds during the seeding period.

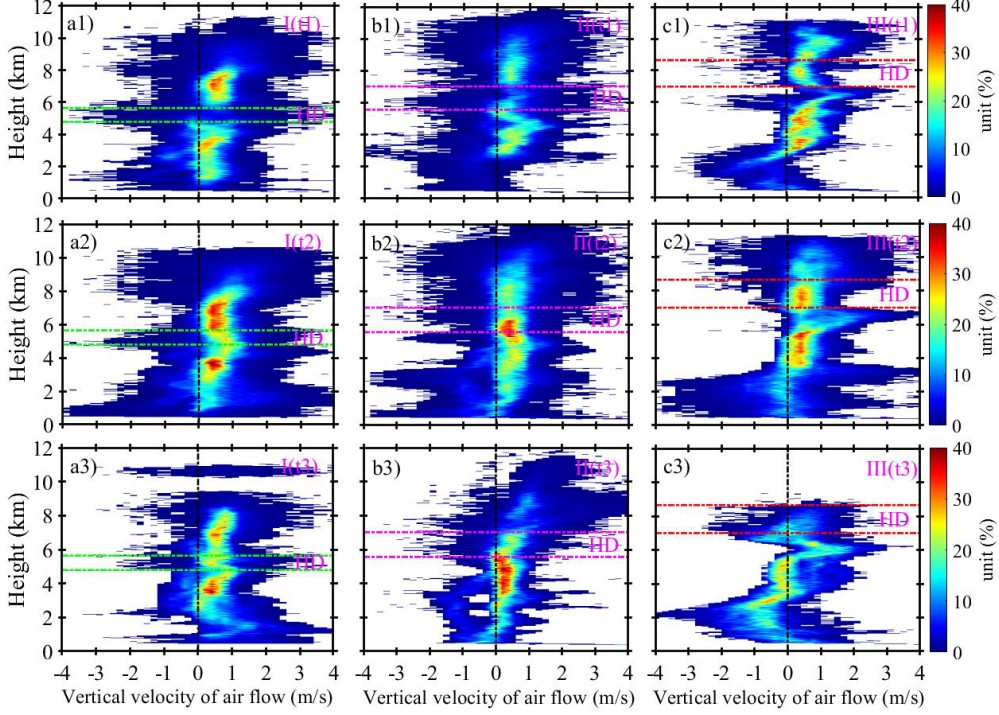


Figure 11 The distribution of probability density with height of the vertical velocity of air flow in the three types of seeder-feeder
process before (t1), during (t2) and after (t3) seeding.
The distribution of vertical velocity airflow in clouds is the reflection of the dynamic structure of clouds. The
airflow in stratiform clouds is usually slow, and the diameter and concentration of cloud particles change little. Fig.
11 shows the DPDHs of the vertical velocity of airflow in the three types of seeding and feeding clouds. It shows
that updraft and downdraft existed simultaneously in the cloud. The vertical velocity of airflow in the upper part of
the seeding clouds was slightly larger than that in the lower part, which transported water vapour needed for the
growth of ice particles, and increased the probability of collision between particles. The updraft velocity at the top
of the feeding clouds was also slightly greater than that at the base. There were the slight difference between the
three types of seeder-feeder processes, among which the type I and II were dominated by weak updrafts before,
during and after seeding, and HD region was also dominated by weak updrafts, the updrafts were mainly distributed
in the range of 0 ~1 ms$^{-1}$ (probability density is greater than 20%). The probability density of strong or weak
updraft (greater than 1 ms$^{-1}$ or less than 0 ms$^{-1}$) was less than 20%. For the type III, before and during seeding, the
DPDHs with height for the vertical velocity of airflow was similar to that of the type I and II, but after seeding, the
large downdraft appeared in the HD region and the middle and lower part of the feeding clouds. Fig. 10c3 also
shows that the cloud particles in the lower part of the feeding clouds mainly moved down, and the reflectivity
factor showed that precipitation appeared at the base of the feeding clouds.

To understand the relationship between cloud particle variation and echo signal, the correlation coefficient

between cloud particle terminal velocity and corresponding reflectivity factor in the three types during seeder-
feeder period (t2) was calculated, and then averaged according to different categories. Based on the height
corresponding to the average HD thickness of the three types of seeder-feeder processes, and the correlation
coefficient profiles and average reflectivity factor profiles of the corresponding categories were obtained as shown
in Fig. 12. The ESD regions were indicated by cyan shaded boxes, the height and thickness of the HD layer in the
three types continuously increased from the type I to the type III, while the thickness of the ESD layer was on the
contrary in Figs. 12. The ESD region in the type I was the thickest and it was the thinnest in the type III, which
gives the conclusion that the HD height was high, and the thickness of ESD region was thin during seeding process.
This process can be understood as that when HD layer is high, the cloud particles are small (that is, light particles in
weight), so their falling speed is also small (see Fig. 10b2), so the depth of their falling into the top layer of the
feeding clouds is also shallow. On the contrary, when the cloud particles in the HD layer are larger (i.e. heavier),
the height of the HD layer is lower, and these particles would fall into the deeper region of the feeding clouds, such
as the type I.

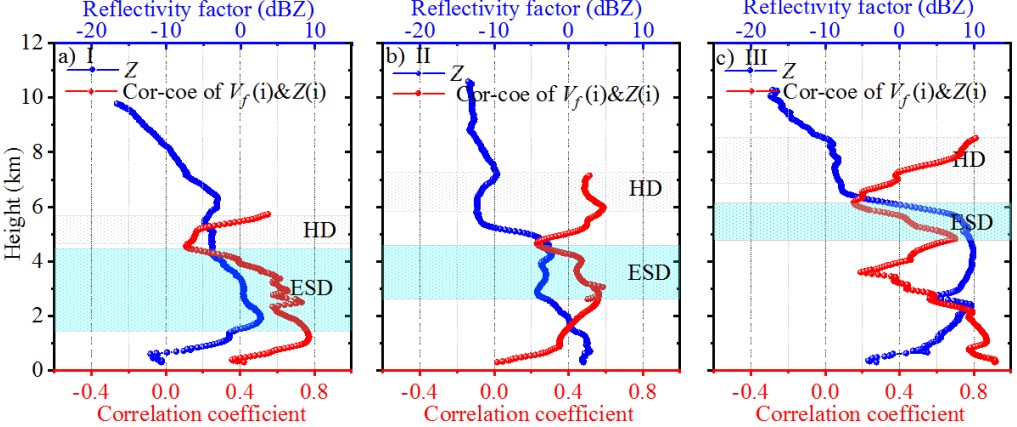


Figure 12   Autocorrelation coefficient profiles (red line) between cloud particle terminal velocity and reflectivity factor (blue line)
during seeding (t2) for the type I (a), the type II (b) and the type III (c)

Fig. 12a shows more details in the feeding clouds such as the reflectivity factor increased with the decrease of

the detection height and reached the maximum value at 2 km, and the correlation coefficient also increased with the
increased reflectivity factor. That indicates close relationship between the reflectivity factor and the particle
terminal velocity. The essence was that when the particle terminal velocity was large, it means that the cloud
particles had the large mass and the large size, and the reflectivity factor must was large. Fig. 10a2 also shows that
there were a certain proportion of cloud particles in the middle and lower part of the ESD region with the large

sinking velocity. However, at the base of feeding clouds, the reflectivity factor and the correlation coefficient decreased, indicating that there was basically no seeding effect at the base of feeding clouds during t2 period. The reflectivity factor increased rapidly but the correlation coefficient decreased rapidly at the top of feeding clouds in the type II. It is estimated that because the seeders in the HD layer just fell into the top of the feeding clouds resulting in the number of cloud particles increased at the top, but these particles did not have time to grow, so although the reflectivity factor increased, the correlation coefficient decreased rapidly. When the seeders drop to a certain depth in the feeding clouds, the interaction between cloud particles such as collision occurs so that the correlation coefficient between particle terminal velocity and reflectivity factor increases synchronously. Below the ESD region, the correlation coefficient decreased rapidly with the decrease of detection height, but the reflectivity factor continued increased, which was probably caused by the large number of particles in the layer. In the type III, as the seeders entered the ESD region, the reflectivity factor increased rapidly with the decreased of detection height together with the correlation coefficient increased rapidly to the maximum. In the detection height was in range of 5 km ~ 3.5 km, the correlation coefficient decreased obviously with the decreasef of height, but the reflectivity factor maintained a large value (about 10 dBZ), which indicated the high concentration of cloud particle in this detection height. In the lower part of the feeding clouds, the reflectivity factor decreased with the decreased of the height, while the correlation coefficient increased, indicating that the particle terminal velocity in this height also decreased. It is likely that the cloud particles were so small that some of them were evaporated, caused both the reflectivity factor and the particle terminal velocity to decrease simultaneously. In general, the depth of seeders fell the feeding clouds was limited, and the lower the height and thinner the HD layer, the lower the height and thicker the thickness of ESD region.

## 5. Conclusions

In this paper, the data of bilayer stratiform cloud in winter to the next spring detected by the MWCR were investigated, and the seeder-feeder process in the bilayer clouds was observed in Xi 'an, China. By defining the key parameters of the seeding and feeding clouds, such as the HD between the bilayer cloud and the ESD of the feeding clouds, the calculation method of DPDH and the analysis method of the correlation coefficient profile between the cloud particle terminal velocity and the reflectivity factor were adopted. The results show that: (1) During the 11 cases of seeder-feeder process in the bilayer cloud, the seeding effect had the significant impact on the macro- and micro- parameters of the feeding clouds, which was mainly manifested in that the seeding effect caused the significant increase of the reflectivity factor and the terminal velocity of cloud particles in the feeding clouds.

Therefore, it was speculated that the seeding effect caused the significant increase in the particle diameter of the
feeding clouds. (2) According to the distribution characteristics of the ESD region thickness and height, the seeder-
feeder processes in bilayer cloud were divided into three types, the type I had thin HD layer with low height, and its
ESD region was thick; The type III had thick HD layer with high height, its ESD region was thin; The values of
both HD and ESD of the type II located at the type I and III. It can be inferred that the lower the height and thinner
the thickness of HD layer, the lower the height and thicker the thickness of ESD region, and vice versa. (3)
According to the analysis results of 11 cases, the related parameter distribution of the seeder-feeder process in
bilayer cloud are shown in Fig. 13, that is, during the evolution of bilayer cloud, the phenomenon of cloud particles
from the lower part of the upper-layer clouds seeded the lower-layer clouds would occur under appropriate weather
background, that is, the distribution of air flow was unique with the height, and there was the relatively obvious
updraft at the top of the seeding clouds. In the seeding layer (the HD layer and ESD region), the downdraft and
cloud particles was larger, and when there brought rainfall, the sinking motion at the base of the feeding clouds was
stronger, and there was a small amount of down-flow region in the seeding and the feeding clouds, but weak
updraft in the bilayer cloud. The seeding process can last up to 2 hours, but most seeding lasts for tens of minutes.
Generally, the seeding occurs at $-25°C \sim -10°C$ in the clouds. The seeding effect plays actions on the precipitation
(rain or snow) intensity in the feeding clouds will be shown in the results of subsequent studies.

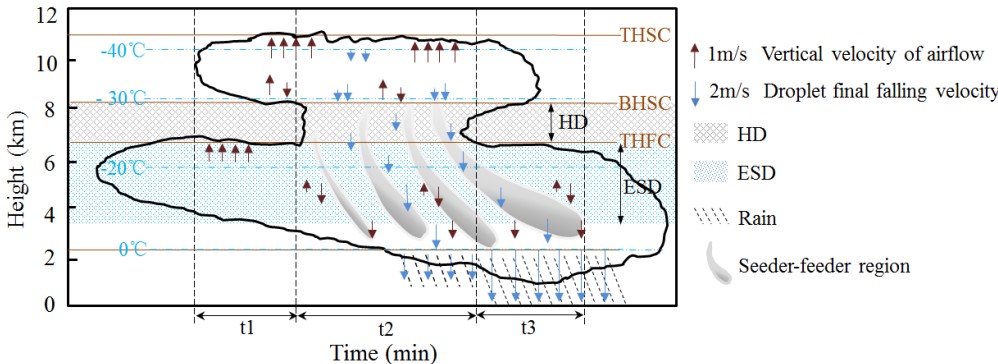


Figure 13. Schematic diagram of the natural seeder-feeder process and related parameter distribution.

## Data availability

The data and codes related to this article are available upon request from the corresponding author.

## Author contributions

Conceptualization: Huige Di
Investigation: Huige Di
Methodology: Huige Di
Software: Yun Yuan
Writing — original draft: Huige Di & Yun Yuan
Writing — review & editing: Huige Di
Supervision: Huige Di
Data collation: Yun Yuan

## Competing interests

The authors declare that they have no conflicts of interest related to this work.

## Acknowledgements

We express our gratitude to the Xi'an Meteorology Bureau of Shaanxi Province, Xi'an, Shehong Li, Shuicheng Bai,
and Mei Cao for providing the relevant supporting data.

## Financial support

This research was supported by the National Natural Science Foundation of China, the Innovative Research Group
Project of the National Natural Science Foundation of China (Grant Nos. 42130612, 41627807), and the Ph.D.
Innovation fund projects of Xi'an University of Technology (Fund No. 310-252072106).

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
