# Peer review of "The characteristics of cloud macro parameters caused by seeder-feeder inside clouds measured by millimeter-wave cloud radar in Xi'an"

_EGUsphere, 2023_

## Referee Comment (RC1)

Based on the observed results of microwave radar, this paper analyzed the seeding process of the upper layer in stable stratus cloud to the lower layer cloud. Specifically, by defining the relevant parameters of the seeding process, the characteristics of these parameters in the seeding process were obtained, such as the seeding depth and the seeding action time. The significance of this paper is to reveal the cloud interaction in vertical direction from the observed results of microwave radar.

The process from cloud to rain is very complex, the upper cloud affected the lower, and produced precipitation, have been widely attention, such as cold cloud precipitation process that involves the upper ice particles falling action on the lower cloud. To this end, the following recommendations are made:

1. Is the method described in the article appropriate for unstable cloud systems?
2. In the article, only one microwave radar was used. If the two radars placed along the direction of cloud movement, is it better for your result?
3. It is very nice for this article that obtained the useful characteristics of seeding process, such as the differences of seeding depth and the seeding action time for the three different seeding processes. My suggestion is that in the follow-up studies by large sample statistics, possibly more specific differences between different seeding types will be obtained.

---

## Community Comment (CC3)

Referee comment

**General comments:**

Natural seeder-feeder process is one of the cloud microphysical processes that have yet been well understood. The authors used observations of bilayer cloud cases collected by MMCR to conduct an analysis on the seeder-feeder mechanism, some parameters are defined to reveal the changes of cloud macro and microphysical characteristics during the seeder-feeder process. Further analysis of the radar moments also gives some interesting results. The manuscript is well organized and easy to follow. However, I have some concerns about the accuracy of retrievals and some technical methods are not well described in the manuscript. Please see my comments below.

**Response:** Thank you very much for your nice comments. Your question and suggestion are very helpful for us to improve the quality of our paper. We appreciate the reviewer's thoughtful review and constructive comments. The following is our point-to-point replies.

**Major comments:**

1. The authors classified the collected cases based on the height difference (HD) and effective seeding depth (ESD) of seeded cloud, both of these parameters need to be calculated from height of cloud top and bottom (either seeding cloud or feeding cloud) but it's not clear how the authors determine them. If the radar echo is used to determine the THSC, BHSC and THFC, then the height is affected by the sensitivity of the MMCR. Further, the authors should also consider how to determine these parameters when the cloud top or cloud bottom is not flat enough.

**Response:**

The heights of cloud top and bottom are determined from radar echo signals. Before determining the two heights, the clutter mixed in signals observed by millimeter wave cloud radar were filtered out. The sensitivity threshold of the radar used in this study is -40dBz, which is sufficient for accurately observing the positions of cloud base and cloud top (Yuan yun,et al. 2023,Atmosphere Measurement Technique).

Usually, the cloud base or cloud top is not flat enough, so reviewer's question for this part is reasonable. However, as our study focuses on stable stratiform clouds, the cloud tops and cloud bases observed in these cases are relatively flat. In this study, THSC is the average height of seed cloud tops during the observation period, BHSC is the average value of the seeder cloud base during the t1 period, and THFC is the average value of the feeder cloud top during the t1 period.

We will supplement the above content in the revised version. Thanks again.

2. The "small particle tracer" method was used to retrieve the vertical air motion, which we all agree is more suitable for stable clouds with small turbulence, I would like to see the authors verify the retrieved vertical air velocity with the radiosonde data in-situ since the authors carried out SP broadening correction.

**Response:**

The opinion of reviewer is right that the "small particle tracer" method is more

suitable for stable clouds with small turbulence. Generally, the results are affected by turbulence, wind shear, and beam width, so the correction is needed. In our study the horizontal wind of sounding data was used to correct the echo power spectrum caused by possible wind shear. However, there is no vertical air velocity in the sounding data, so it is not possible to verify the retrieved vertical air velocity with the radiosonde data in-situ. Perhaps we can do it in the future by other way. Thanks your nice question.

3. In section 4, the authors set the HD threshold as 1 km for classification, is there any statistical analysis performed to determine the threshold?

**Response:**

The HD threshold in this study is obtained based on the statistical results of the 11 cases listed in Table 4. Thanks.

**Minor comments:**

1. I understand the authors use "final particle falling velocity" refer to the particle falling velocity in static air, but it's not appropriate to say "final" velocity, "particle terminal velocity" or "particle falling velocity" is encouraged to use.

**Response:**

Thank you for the suggestions. In the revision, we will revise from "final particle falling velocity" to "particle terminal velocity".

2. P3L61: "retrieval" to retrieve

**Response:** OK, thank you very much.

3. The use of "observational data" is in question.

**Response:** Thanks. Now it is modified as "observation data".

4. P15L380: I guess the authors means both seeder and feeding clouds?

**Response:**

Thanks your nice reminding. P15 L380: "seeded and seeded cloud increase during the seeding period" should be "seeding and feeding cloud increase during the seeding period".

---

## Community Comment (CC5)

General comments:

The manuscript presents analysis of cloud and precipitation development related to the seeder-feeder mechanism, based on ground-based Ka-band millimeter-wave cloud radar (MMCR) and microwave radiometer (MWR) in Xi'an, China. Through analysis of the reflectivity factor and the radial velocity of cloud particles detected by MMCR, and the retrieved cloud dynamics parameters (vertical velocity of airflow and falling velocity of cloud particles), the authors showed that the reflectivity factor in the cloud are significantly enhanced during the seeder-feeder period, and the magnitude of the enhancements are different for clouds with different HD (the height difference between the seeding cloud base and the feeding cloud top) and ESD (the effective seeding depth). While the manuscript is understandable as a whole, many phrases and explanations are not clear or even incorrect. I think the quality of the manuscript needs a significant improvement.

**Response:** Thank you very much for your nice comments. Your question and suggestion are very helpful for us to improve the quality of our paper. We appreciate the reviewer's thoughtful review and constructive comments. The following is our point-to-point replies.

Specific Comments:

1.  When explaining the seeder-feeder mechanism (Lines 31-33), the authors said: When these seeders meet lower cloud droplets with ice phase or in supercool water state, the droplets will grow larger by riming or vapor deposition via the Wegener-Bergeron-Findeisen effect (Bergeron 1935; He et al., 2022). This is conceptually incorrect. It should be the ice particles that grow in the WBF process, not the droplets.

    **Answer:** Thank you very much for your nice reminding. We went through the references, and you were right. When ice crystals meet lower cloud droplets with ice phase or in supercool water state, they grow by riming or vapor deposition via the Wegener-Bergeron-Findeisen effect (Bergeron 1935; He et al., 2022; Ulrike Proske et al., 2021).

2.  Probably the most significant contribution of this article is by defining the height

difference between the seeding cloud base and the feeding cloud top (HD) and the effective seeding depth (ESD), the authors found that the seeding effect are different for cloud with different HD. However, the similar expressions "The lower and thinner the HD height was, the lower and thicker the ESD height was. On the contrary, the higher the HD height, the higher and thinner the ESD height" appeared in the Abstract (Lines 20-21) and in the Conclusions (Lines 529-530) are hard to understand. The heights of HD and EST cannot be described as thinner or thicker.

**Answer:** Thank you very much for your nice reminding. The abstract (Lines 529-530) and conclusion (Lines 529-530) have been revised as follows:

Abstract(Lines 20-21):The lower the height and thinner the thickness of the HD, the lower the height and thicker the thickness of ESD; On the contrary, the higher the height and the thicker the thickness of the HD, the higher the height and the thinner the thickness of the ESD.

Conclusion(Lines 529-530):The lower the height and thinner the thickness of HD, the lower the height and thicker the thickness of ESD.

More detailed comments:

1. The title may be changed as "The characteristics of cloud macro parameters caused by seeder-feeder process inside clouds measured by millimeter-wave cloud radar in Xi'an, China";

 **Answer:** Thanks. The title has been changed based on your suggestions.

2. Line 13: Change "China Xi'an" to "Xi'an, China";

**Answer**: Thanks. It has been changed into "Xi'an, China".

3. Lines 18-19: "But there are different enhancements among the reflectivity factor profiles for the three seeding-feeding processes" may be modified to "But the magnitudes of the enhancements among the three seeding-feeding processes are different" to avoid duplication with the previous sentence;

**Answer**: Thanks. That has been modified.

4. Line 26: "lead" may be modified to "promote";

**Answer**: Thanks. That has been modified.

5.Lines 29-30 are not clear;

**Answer**: The sentence has been rewritten as follows: "The seeder-feeder process is a phenomenon that ice crystals as seeder, from an upper cloud fall into a lower cloud or a lower-lying part of the same cloud, which is either liquid, ice or mixed phase." Thanks.

6. Lines 34-35: There are syntax errors in this expression;

**Answer**: Thank you very much, the sentence has been rewritten as follows: "Therefore, it is important to understand the seeder-feeder mechanism, which can be helpful to improve the representation of cloud processes in weather and climate models, and weather forecasts of precipitation, and ultimately to reduce uncertainty in climate simulations. "

7. Lines 36-38: The sentence is incomplete or not clear;

**Answer**: The sentence has been rewritten as follows: "The seeding-feeding phenomenon has been studied by observations and simulations in operations of the artificial precipitation enhancement, and it was found that the distinct changes in both cloud and precipitation properties. (French et al., 2018; Ramelli et al., 2021; Dong et al., 2021)"
Thanks.

8. Lines 42-44: There is logical problem in this sentence;

**Answer**: The sentence has been rewritten as "In the 1980s in China, Hong et al., (2011 and 2012) established a cloud model that simulated the formation of stratiform clouds. In the model, the seeding-feeding process was emphasized."

9. Lines51-54: The expression is not clear to me. What scientific questions are still existing related to the seeder-feeder process?

**Answer**: This paragraph has been revised into "The microphysical parameters of the seederfeeder process appeared within mixed-phase clouds have been investigated by using the ground-based remote sensing instruments (Ramelli et al., 2021). However, there is still a lack of the specific characteristics, such as the height difference between the seeding cloud base and the feeding cloud top (HD) and the effective seeding depth (ESD), to represent the feature of the seeder-feeder process. In the meantime, the characteristic of air vertical motion, particle terminal velocity inside cloud during seeding-feeding process is still poorly understood."

10. Line 57: Why several cloud layers are needed?

**Answer**: Because our study focuses on the external "seeding-feeding" process that occurs between double-layer clouds, rather than the effect of the upper layer on the lower layer within the same cloud body. Thanks.

11. Line 64: "the seeder-feeder processes" or "the seeder-feeder process"?

**Answer**: It has been changed into "the seeder-feeder process". Thanks.

12. The sentence in Lines 66-67: "The parameters of microphysics, dynamics and thermodynamics during the seeder-feeder process were focused on analysis" is not clear to me.

**Answer**: This sentence has been deleted in the revised manuscript. Thanks.

13. Line 83: "observations data" may be changed to "observational data".

**Answer**: OK, Thanks. "observations data" has been changed to "observation data".

14. Line 109: "which reduces the trouble of eliminating terrain clutter in observation data quality control", needs to be rewritten.

**Answer**: OK, Thanks. It is now rewritten as "which reduces errors of the terrain clutter in observational data"

15. What does "air haze" in Line 110 mean?

**Answer**:It means aerosol. So, it has been modified to "aerosol". Thanks.

16. Lines 109-113 are not clear.

**Answer**:This paragraph has been revised into "The echo signals of floating debris in the low-level atmosphere have the characteristics of a small reflectivity factor, low velocity, and large spectral width. To further eliminate interfering wave information, we obtained the data quality control threshold by counting the characteristic changes in planktonic echoes in the boundary layer under cloud free conditions (Yuan et al., 2022)." Thanks.

17. Lines 120-121: the expression "Because the descending the cloud particles velocity in different phase states is different, the influence on the vertical velocity of the airflow in the cloud is different" needs to be written.

**Answer**:Now, the sentence has been rewritten as "The falling speed of cloud particles varies due to the influence of phase state, which in turn affects the magnitude of vertical airflow velocity." Thanks.

18. Line 128: What does mean by "In the identified I supercooled water region"?

**Answer**:That is a clerical error. It has been modified as "In the identified supercooled water region". Thanks.

19. Line 129: "When there was a drizzle, the SP of MMCR usually shows a bimodal distribution": a drizzle?

**Answer**:It has been modified as ""When it drizzles, the SP of MMCR usually shows the bimodal distribution." Thanks.

20. Line 138: change "this season" to "these seasons";

**Answer**:OK, thank you very much.

21. Line 141: "cloud bottom" should be changed as "cloud base" , similar changes should also be made in other places;

**Answer**:OK, thank you very much.

22. Lines 139-141: How the heights of cloud top and cloud base are defined?

**Answer**:It is easy to identify cloud base and cloud top based on the echo profiles measured by millimeter wave cloud radar, which has been descripted in the reference published in AMT (Yuan yun, 2021 AMT). The heights of cloud top and cloud base are usually clear in the echo profiles. Thanks.

23. Lines 152-153: "······ seeding time t2 at 98.2min, and feeding cloud development duration at more than 2hr 30min" needs rewritten.

**Answer**:The sentence has been rewritten as "It also shown that the bilayer cloud is stable during this period, with THSC stable at 8 km, BHSC at 5.5 km, THFC at 4.2 km, DH at 0.85 km. The seeding process lasts for about 98.2 minutes (t2), and feeding cloud development duration reaches more than 2 hours and 30 minutes" . Thanks.

24. Line 166 and several other places: "the final falling velocity" is "the terminal fall velocity"?

**Answer**:OK, in the revised manuscript, "the final falling velocity" has been changed to "the particle terminal velocity". Thanks.

25. Lines 170-171: why the unit of width of the velocity spectrum is m/s?

**Answer**:The following is our description. the width of the velocity spectrum is the degree to which the Doppler velocity of particles within a radar's effective detection volume deviates from its average value. The larger the difference in particle size, the greater the corresponding difference in falling end velocity, resulting in a wider Doppler velocity spectrum. The formula 1 is the definition of the width of the velocity spectrum, so the unit of width of the velocity spectrum is m/s. Thanks.

$$SP = \sqrt{\frac{\sum_{i=V_L}^{V_R}(i-V_r)^2 \times (S_i - P_N)}{\sum_{i=V_L}^{V_R}(S_i - P_N)}} \tag{1}$$

Here, $V_L$: Doppler velocity at the left endpoint of the Doppler spectra (unit: m/s); $V_R$: Doppler velocity at the right endpoint of the Doppler spectra (unit: m/s); $S_i$: cloud signals (unit: dBm); $P_N$: noise level (unit: dBm);

26. Lines 184-186: "But in some altitudes, there are the airflow sinking movements, which can be explained the needs of airflow sinking movement short-term to achieve mass balance": What does this mean?

  **Answer**:For easy comprehension, the sentence has been rewritten as "There is rarely a large-scale and prolonged air sinking and rising movement in the seeding cloud and feeding cloud, but alternating upward and downward movements occur. ". Thanks.

27. Line 187: "the sinking velocity of the cloud particles is in the range of –1~ –4 ms$^{-1}$ during seeding process": What size cloud particles can get such falling velocity? The sinking velocity is the terminal fall velocity?

  **Answer:** According to Fig. 3 in the manuscript, the terminal velocity of particles in the seeding and feeding area is from -1 to -4 m/s, but most of them are less than -2.5 m/s. According to the cloud phase in Fig. 4, the particles are snowflakes in the cloud seeding and feeding area. The particle size is related to the shape of snowflakes and the final falling velocity, so it difficult to accurately quantify particle size. Table 1 lists the relationship between snowflake diameter and terminal velocity.

Table 1  the relations of cloud particles and diameter   (Tao R., et al. TGRS, 2020)

| Study | Relation | Snow Type |
| --- | --- | --- |
| Brandes (2002) [22] | $v_t = -0.1021 + 4.932D - 0.9551D^2 + 0.07934D^3 - 0.002362D^4$ | Rain |
| Jeong-Eun (2015) [23] | $v = 1.03D^{0.71}$ | Needle |
| Locatelli and Hobbs (1974) [24] | $v = 1.3D^{0.66}$ | Lump graupel |
|  | $v = 0.79D^{0.27}$ | Densely rimed aggregates |
|  | $v = 0.81D^{0.16}$ | Unrimed aggregates |
| This study | $v = 1.03D^{0.25}$ | low terminal velocity cases |
|  | $v = 1.29D^{0.29}$ | high terminal velocity cases |

According to the speculation from table 1, the size of the snow particles in the cloud is distributed between 1mm and 6mm, and most of them are below 3mm.

 Yes, the sinking velocity is the particle terminal velocity. Thanks.

28. Lines 207-210: "During being seeded, ice particles were the main component in the cloud. After being seeded, the ice particles in the lower part of the feeding cloud lasted for a long time (maintaining the whole t3 period), while the supercooled water layer at the top was obvious": This needs more explanation.

**Answer:**

"Before seeding, the larger downward mean Doppler velocity (Figure 2b) was detected in the lower part of the seeding cloud, which indicates that the cloud process has transformed from ice to snow with large particle sizes. Snowflakes, as seeders, fall into the mixed phase cloud containing supercooled water, so that the Wegener-Bergeron-Findeisen effect occurred. That effect causes the mixed phase cloud to rapidly transform into ice. Because it takes time for particles to fall, so the seeding will continue to the middle and lower parts of the feeding clouds, and snow keeps for a long time (maintaining the entire t3 period); In the top region of the unaffected feeding cloud, the cloud phase remains supercooled water, which is consistent with the observation results in Shupe (2007)." Thanks.

29. Lines 211-212: "The instantaneous water vapor flux structure (Fig. 4b) indicates that the seeding cloud is smaller than the feeding cloud": smaller in size or water vapor flux?

**Answer:** the sentence has been rewritten as "From Figure 4, it can be seen that the instantaneous water vapor flux of the seeding cloud is smaller than that of the feeding cloud." Thanks.

30. "seeing-feeding" in Lines 213-214 should be "seeding-feeding";

**Answer:** OK, thank you very much.

31. Lines 229-233: not clear;

**Answer**: This paragraph has been revised as "The cloud particle with larger diameter has a larger falling speed under the action of gravity. In order to reveal the relationship between particle size and echo signal in the process of seeder and feeder. The statistical classification method of equal samples is adopted to find the relationship. All signal values (echo reflectivity,

radial velocity, spectral width, particle falling velocity, and vertical airflow velocity) are reordered according to their corresponding echo reflectivity values from small to large, and then compared in the equal sample." Thanks.

32. Line 236: Change "Following to this principle" to "Following this principle";

**Answer**:OK, thank you very much.

33. Lines 240-241: "⋯⋯the corresponding average profile of cloud particle parameter profile for the three intensity echoes is also gained" needs rewritten;

**Answer**:It has been rewritten as "⋯⋯ the corresponding average profile of cloud particle parameter for the three intensity echoes is also obtained." Thanks.

34. Lines 287-288: "⋯⋯ after the seeding, the cloud particle size distribution and particle velocity of the bilayer cloud reach a relatively balanced and stable state through complex thermodynamic and dynamic interactions": needs more physical explanation.

**Answer**:Due to the fact that echo reflectivity factor, radial velocity, and falling terminal velocity reflect particle size, and spectral width reflects particle size distribution and particle category. in the end of seeding, the cloud particle size distribution and particle velocity of the bilayer cloud may reach a relatively balanced and stable state through the complex microphysical and dynamic interactions in the t2 period." Thanks.

35. Lines 310-312: "Therefore, the Effective Seeding Depth (ESD) is defined as the height difference between the top height of the feeding cloud and from the height down to the height of the maximum correlation coefficient, which represents the influence of seeders on the seeding cloud": I do not understand.

**Answer**:We confused. The last three words are " the feeding cloud" in the sentence. In the revised manuscript the error has been modified. Thanks.

36. What does it mean by "sowing effect" in Line 315?

**Answer**:That is a clerical error. "sowing effect" has been changed into "seeding effect" in Line 315. Thanks.

37. There are more expression problems in the remaining part of the text. I suggest the quality of the whole text to be carefully checked and revised.

**Answer**:Thanks. We have checked and revised manuscript sentence by sentence to avoid mistakes.

---

## Author Response (AR1)

**Response to Editor**

**Dear Editor & Prof.**

   We greatly thank you and the reviewers for the thorough and valuable suggestions to our work, and thank the valuable comments and suggestions by peer experts in the open discussion. We have made a point-to-point response to these opinions and suggestions, and all the comments have been addressed in the revised manuscript. We believe that the quality of the manuscript has been promoted now. The responses to each comment are given below.

   The modifications we have made to this manuscript have been indicated in blue font in the Author's track-changes file (we have found that the Author's track-changes can only be used in PDF, not in Word in Upload stage).

   Thank you very much for considering our work!

   Yours sincerely,

   Huige Di and Yun Yuan
   Xi'an University of Technology
   dihuige@xaut.edu.cn

**Response to quick reports from Anonymous Referee #1**

By defining the parameters of the seeding depth and seeding time of the upper cloud on the lower cloud, this paper reveals the seeding phenomenon and process characteristics of the stable double layer cloud.

Its innovation is mainly manifested in defining relevant parameters describing the seeding process for the first time, such as the seeding depth parameter, and classifying the seeding process according to the seeding depth and its height, which lays a foundation for further study on the seeding mechanism among clouds. In the aspect of analysis, the analysis method of correlation coefficient profile is used for the first time to expose the relationship between cloud particle final velocity and cloud reflectivity factor, and thus reveal the essence behind the seeding phenomenon.
**Response:** Thank you very much for your nice comments.

It is hoped that further study will be carried out on how seeding depth and seeding time affect the development characteristics of the lower layer cloud, such as the relationship between the development time and seeding amount of the lower layer cloud after seeding.
**Response:** Thank you very much for your nice comments. The seeding effect plays actions on the precipitation (rain or snow) intensity in the feeding cloud will be studied in the future.

**Response to quick reports from Anonymous Referee #2**

**General comments:**
Natural seeder-feeder process is one of the cloud microphysical processes that have yet been well understood. The authors used observations of bilayer cloud cases collected by MMCR to conduct an analysis on the seeder-feeder mechanism, some parameters are defined to reveal the changes of cloud macro and microphysical characteristics during the seeder-feeder process. Further analysis of the radar moments also gives some interesting results. The manuscript is well organized and easy to follow. However, I have some concerns about the accuracy of retrievals and some technical methods are not well described in the manuscript. Please see my comments below.
**Response:** Thank you very much for your nice comments. Your question and suggestion are very helpful for us to improve the quality of our paper. We appreciate the reviewer's thoughtful review and constructive comments. The following is our point-to-point replies.

**Major comments:**
1. The authors classified the collected cases based on the height difference (HD) and effective seeding depth (ESD) of seeded cloud, both of these parameters need to be calculated from height of cloud top and bottom (either seeding cloud or feeding cloud) but it's not clear how the authors determine them. If the radar echo is used to determine the THSC, BHSC and THFC, then the height is affected by the sensitivity of the MMCR. Further, the authors should also consider how to determine these parameters when the cloud top or cloud bottom is not flat enough.
**Response:**
The heights of cloud top and bottom are determined from radar echo signals. Before determining the two heights, the clutter mixed in signals observed by millimeter wave cloud radar were filtered out. The sensitivity threshold of the radar used in this study is -40dBz, which is sufficient for accurately observing the positions of cloud base and cloud top (Yuan yun,et al. 2023,Atmosphere Measurement Technique).

Usually, the cloud base or cloud top is not flat enough, so reviewer's question for this part is reasonable. However, as our study focuses on stable stratiform clouds, the cloud tops and cloud bases observed in these cases are relatively flat. In this study, THSC is the average height of seed cloud tops during the observation period, BHSC is the average value of the seeder cloud base during the t1 period, and THFC is the average value of the feeder cloud top during the t1 period.

We will supplement the above content in the revised version. Thanks again.

2. The "small particle tracer" method was used to retrieve the vertical air motion, which we all agree is more suitable for stable clouds with small turbulence, I would like to see the authors verify the retrieved vertical air velocity with the radiosonde data in-situ since the authors carried out SP broadening correction.

**Response:**

The opinion of reviewer is right that the "small particle tracer" method is more suitable for stable clouds with small turbulence. Generally, the results are affected by turbulence, wind shear, and beam width, so the correction is needed. In our study the horizontal wind of sounding data was used to correct the echo power spectrum caused by possible wind shear. However, there is no vertical air velocity in the sounding data, so it is not possible to verify the retrieved vertical air velocity with the radiosonde data in-situ. Perhaps we can do it in the future by other way. Thanks, your nice question.

3. In section 4, the authors set the HD threshold as 1 km for classification, is there any statistical analysis performed to determine the threshold?

**Response:**

The HD threshold in this study is obtained based on the statistical results of the 11 cases listed in Table 4. Thanks.

**Minor comments:**

1. I understand the authors use "final particle falling velocity" refer to the particle falling velocity in static air, but it's not appropriate to say "final" velocity, "particle terminal velocity" or "particle falling velocity" is encouraged to use.

**Response:**

Thank you for the suggestions. In the revised manuscript, we will revise from "final particle falling velocity" to "particle terminal velocity".

2. P3L61: "retrieval" to retrieve

**Response:** OK, thank you very much.

3. The use of "observational data" is in question.

**Response:** Thanks. Now it is modified as "observation data".

4. P15L380: I guess the authors means both seeder and feeding clouds?

**Response:**

Thanks your nice reminding. P15 L380: "seeded and seeded cloud increase during the seeding period" should be "seeding and feeding cloud increase during the seeding period".

**Response to reports from Anonymous Referee #1**

**General comments:**
Based on the observed results of microwave radar, this paper analyzed the seeding process of the upper layer in stable stratus cloud to the lower layer cloud. Specifically, by defining the relevant parameters of the seeding process, the characteristics of these parameters in the seeding process were obtained, such as the seeding depth and the seeding action time. The significance of this paper is to reveal the cloud interaction in vertical direction from the observed results of microwave radar.

   The process from cloud to rain is very complex, the upper cloud affected the lower, and produced precipitation, have been widely attention, such as cold cloud precipitation process that involves the upper ice particles falling action on the lower cloud.

**Response:** We appreciate the reviewer's thoughtful review and constructive comments.

**Specific comments:**
   1. Is the method described in the article appropriate for unstable cloud systems?

   **Response:** In this study, the method of fixed-base observation using one radar is suitable for stable stratiform cloud systems. In the unstable convective cloud system with change rapidly, the corresponding method will be used to deal with it. Our idea of this paper is still useful.

   2. In the article, only one microwave radar was used. If the two radars placed along the direction of cloud movement, is it better for your result?

   **Response:** Although a single radar is used in this study, the spatiotemporal conversion method in this paper is still valid and reliable for the stable cloud system. For unstable cloud systems, such as convective cloud systems, if multiple radars can be used to observe at the same time, the results will be more accurate and convincing.

   3. It is very nice for this article that obtained the useful characteristics of seeding process, such as the differences of seeding depth and the seeding action time for the three different seeding processes. My suggestion is that in the follow-up studies by large sample statistics, possibly more specific differences between different seeding types will be obtained.

   **Response:** In this paper, the seeding effect between double-layer cloud systems is analyzed in terms of macro characteristics. In the future, the rainfall of double-layer cloud systems and mixed phase cloud systems in this region will be studied and analyzed in detail, in order to expose the rainfall mechanism in this region.

**Response to Aoqi Zhang**

This paper investigated the 'seeder-feeder' process of bilayer stratiform clouds. Using ground-based MMCR and MWR) in spring and autumn from 2020 to 2022 in Xi'an, the authors defined several key parameters of 'seeder-feeder' process, including Height Difference (HD) of two clouds and effective seeding depth (ESD), and studied the relationship of the key parameters. The idea of this paper is innovative, and presents well references for investigating 'seed-feeder' processes and effects in other regions. The studies are well organized with clear results.

**Response :**

Thank you very much for your comment of our paper, and your question and suggestion are very helpful for us to improve the quality of the paper.

The following is our replies to your question.

Specifically, I'm very curious about one question: how does the 'seeder-feeder' process initiate (end time of t1 and start time of t2)? It would be nice if the authors discuss more clearly about this issue.

**Answer:**

The starting time, t2, is determined based on the distribution of the echo reflectivity factor, which is the time when the reflectivity factor of the upper and lower clouds begins firstly to connect (see Figure 2 in the text). At this time, particles in the upper cloud fall into the lower cloud.

In this study, we marked the start time, t2, by using our eye observation method. In the future, if more statistical and business research is conducted, we will develop algorithms to automatically recognize the start time t2.

Line 388:delete 'f' after 'The'

**Answer:**

Thanks, it has been deleted in the revised manuscript.

**Response to Yan Yin**

**General comments:**

The manuscript presents analysis of cloud and precipitation development related to the seeder-feeder mechanism, based on ground-based Ka-band millimeter-wave cloud radar (MMCR) and microwave radiometer (MWR) in Xi'an, China. Through analysis of the reflectivity factor and the radial velocity of cloud particles detected by MMCR, and the retrieved cloud dynamics parameters (vertical velocity of airflow and falling velocity of cloud particles), the authors showed that the reflectivity factor in the cloud are significantly enhanced during the seeder-feeder period, and the magnitude of the enhancements are different for clouds with different HD (the height difference between the seeding cloud base and the feeding cloud top) and ESD (the effective seeding depth). While the manuscript is understandable as a whole, many phrases and explanations are not clear or even incorrect. I think the quality of the manuscript needs a significant improvement.

**Response:** Thank you very much for your nice comments. Your question and suggestion are very helpful for us to improve the quality of our paper. We appreciate the reviewer's thoughtful review and constructive comments. The following is our point-to-point replies.

Specific Comments:

1.  When explaining the seeder-feeder mechanism (Lines 31-33), the authors said:

When these seeders meet lower cloud droplets with ice phase or in supercool water state, the droplets will grow larger by riming or vapor deposition via the Wegener-Bergeron-Findeisen effect (Bergeron 1935; He et al., 2022). This is conceptually incorrect. It should be the ice particles that grow in the WBF process, not the droplets.

**Answer:** Thank you very much for your nice reminding. We went through the references, and you were right. When ice crystals meet lower cloud droplets with ice

phase or in supercool water state, they grow by riming or vapor deposition via the Wegener-Bergeron-Findeisen effect (Bergeron 1935; He et al., 2022; Ulrike Proske et al., 2021).

2.  Probably the most significant contribution of this article is by defining the height difference between the seeding cloud base and the feeding cloud top (HD) and the effective seeding depth (ESD), the authors found that the seeding effect are different for cloud with different HD. However, the similar expressions "The lower and thinner the HD height was, the lower and thicker the ESD height was. On the contrary, the higher the HD height, the higher and thinner the ESD height" appeared in the Abstract (Lines 20-21) and in the Conclusions (Lines 529-530) are hard to understand. The heights of HD and EST cannot be described as thinner or thicker.

**Answer:** Thank you very much for your nice reminding. The abstract (Lines 529-530) and conclusion (Lines 529-530) have been revised as follows:

Abstract(Lines 20-21): The lower the height and thinner the thickness of the HD, the lower the height and thicker the thickness of ESD; On the contrary, the higher the height and the thicker the thickness of the HD, the higher the height and the thinner the thickness of the ESD.

Conclusion(Lines 529-530): The lower the height and thinner the thickness of HD, the lower the height and thicker the thickness of ESD.

More detailed comments:
1. The title may be changed as "The characteristics of cloud macro parameters caused by seeder-feeder process inside clouds measured by millimeter-wave cloud radar in Xi'an, China";

 **Answer:** Thanks. The title has been changed based on your suggestions.

2. Line 13: Change "China Xi'an" to "Xi'an, China";

**Answer**:Thanks. It has been changed into "Xi'an, China".

3.  Lines 18-19: "But there are different enhancements among the reflectivity factor profiles for the three seeding-feeding processes" may be modified to "But the magnitudes of the enhancements among the three seeding-feeding processes are different" to avoid duplication with the previous sentence;

**Answer**:Thanks. That has been modified.

4. Line 26: "lead" may be modified to "promote";

**Answer**:Thanks. That has been modified.

5.Lines 29-30 are not clear;

**Answer**:The sentence has been rewritten as follows: "The seeder-feeder process is a phenomenon that ice crystals as seeder,  from an upper cloud fall into a lower cloud or a lower-lying part of the same cloud , which is either liquid, ice or mixed phase. " Thanks.

6. Lines 34-35: There are syntax errors in this expression;

**Answer**:Thank you very much, the sentence has been rewritten as follows: "Therefore, it is important to understand the seeder-feeder mechanism, which can be helpful to improve the representation of cloud processes in weather and climate models, and weather forecasts of precipitation, and ultimately to reduce uncertainty in climate simulations. "

7. Lines 36-38: The sentence is incomplete or not clear;

**Answer**:The sentence has been rewritten as follows:  "The seeding-feeding phenomenon has been studied by observations and simulations in  operations of the artificial precipitation enhancement, and it was found that the distinct changes in both cloud and precipitation properties. (French et al., 2018; Ramelli et al., 2021; Dong et al., 2021)"
Thanks.

8. Lines 42-44: There is logical problem in this sentence;

**Answer**:The sentence has been rewritten as "In the 1980s in China, Hong et al., (2011 and 2012) established a cloud model that simulated the formation of stratiform clouds. In the model, the seeding-feeding process was emphasized."

9. Lines51-54: The expression is not clear to me. What scientific questions are still existing related to the seeder-feeder process?

**Answer**:This paragraph has been revised into "The microphysical parameters of the seeder-feeder process appeared within mixed-phase clouds have been investigated by using the ground-based remote sensing instruments (Ramelli et al., 2021). However, there is still a lack of the specific characteristics, such as the height difference between the seeding cloud base and the feeding cloud top (HD) and the effective seeding depth (ESD), to represent the feature of the seeder-feeder process. In the

meantime, the characteristic of air vertical motion, particle terminal velocity inside cloud during seeding-feeding process is still poorly understood."

10. Line 57: Why several cloud layers are needed?

**Answer**:Because our study focuses on the external "seeding-feeding" process that occurs between double-layer clouds, rather than the effect of the upper layer on the lower layer within the same cloud body.  Thanks.

11. Line 64: "the seeder-feeder processes" or "the seeder-feeder process"?

**Answer**: It has been changed into "the seeder-feeder process".  Thanks.

12. The sentence in Lines 66-67: "The parameters of microphysics, dynamics and thermodynamics during the seeder-feeder process were focused on analysis" is not clear to me.

**Answer**: This sentence has been deleted in the revised manuscript.  Thanks.

13. Line 83: "observations data" may be changed to "observational data".

**Answer**:OK, Thanks.  "observations data" has been changed to "observation data".

14. Line 109: "which reduces the trouble of eliminating terrain clutter in observation data quality control", needs to be rewritten.

**Answer**: OK, Thanks.  It is now rewritten as "which reduces errors of the terrain clutter in observational data"

15. What does "air haze" in Line 110 mean?

**Answer**: It means aerosol. So, it has been modified to "aerosol". Thanks.

16. Lines 109-113 are not clear.

**Answer**:This paragraph has been revised into "The echo signals of floating debris in the low-level atmosphere have the characteristics of a small reflectivity factor, low velocity, and large spectral width. To further eliminate interfering wave information, we obtained the data quality control threshold by counting the characteristic changes in planktonic echoes in the boundary layer under cloud free conditions (Yuan et al., 2022). "  Thanks.

17. Lines 120-121: the expression "Because the descending the cloud particles velocity in different phase states is different, the influence on the vertical velocity of

the airflow in the cloud is different" needs to be written.

**Answer** : Now, the sentence has been rewritten as "The falling speed of cloud particles varies due to the influence of phase state, which in turn affects the magnitude of vertical airflow velocity."  Thanks.

18. Line 128: What does mean by "In the identified l supercooled water region"?

**Answer** : That is a clerical error. It has been modified as "In the identified supercooled water region". Thanks.

19. Line 129: "When there was a drizzle, the SP of MMCR usually shows a bimodal distribution": a drizzle?

**Answer** :  It has been modified as ""When it drizzles, the SP of MMCR usually shows the bimodal distribution."  Thanks.

20. Line 138: change "this season" to "these seasons";

**Answer** : OK, thank you very much.

21. Line 141: "cloud bottom" should be changed as "cloud base" , similar changes should also be made in other places;

**Answer** :  OK, thank you very much.

22. Lines 139-141: How the heights of cloud top and cloud base are defined?

**Answer** : It is easy to identify cloud base and cloud top based on the echo profiles measured by millimeter wave cloud radar, which has been descripted in the reference published in AMT (Yuan yun, 2021 AMT).  The heights of cloud top and cloud base are usually clear in the echo profiles.  Thanks.

23. Lines 152-153: "…… seeding time t2 at 98.2min, and feeding cloud development duration at more than 2hr 30min" needs rewritten.

**Answer** :  The sentence has been rewritten as "It also shown that the bilayer cloud is stable during this period, with THSC stable at 8 km, BHSC at 5.5 km, THFC at 4.2 km, DH at 0.85 km. The seeding process lasts for about 98.2 minutes (t2), and feeding cloud development duration reaches more than 2 hours and 30 minutes" . Thanks.

24. Line 166 and several other places: "the final falling velocity" is "the terminal fall velocity"?

**Answer** : OK, in the revised manuscript, "the final falling velocity" has been

changed to "the particle terminal velocity". Thanks.

25. Lines 170-171: why the unit of width of the velocity spectrum is m/s?

**Answer** :The following is our description. the width of the velocity spectrum is the degree to which the Doppler velocity of particles within a radar's effective detection volume deviates from its average value. The larger the difference in particle size, the greater the corresponding difference in falling end velocity, resulting in a wider Doppler velocity spectrum. The formula 1 is the definition of the width of the velocity spectrum, so the unit of spectrum width is m/s.  Thanks.

$$\sigma_v = \sqrt{\frac{\sum_{i=V_L}^{V_R}\left(i-V_r\right)^2 \times \left(S_i - P_N\right)}{\sum_{i=V_L}^{V_R}\left(S_i - P_N\right)}} \tag{1}$$

Here, $V_{L:}$ : Doppler velocity at the left endpoint of the Doppler spectra (unit: m/s) ;

$V_R$ : Doppler velocity at the right endpoint of the Doppler spectra (unit: m/s) ; $S_i$ : cloud signals (unit:dBm) ; $P_N$ : noise level (unit:dBm) ;

26. Lines 184-186: "But in some altitudes, there are the airflow sinking movements, which can be explained the needs of airflow sinking movement short-term to achieve mass balance": What does this mean?

 **Answer** :For easy comprehension, the sentence has been rewritten as "There is rarely a large-scale and prolonged air sinking and rising movement in the seeding cloud and feeding cloud, but alternating upward and downward movements occur. ". Thanks.

27. Line 187: "the sinking velocity of the cloud particles is in the range of −1~ −4 ms⁻¹ during seeding process": What size cloud particles can get such falling velocity? The sinking velocity is the terminal fall velocity?

 **Answer:** According to Fig. 3 in the manuscript, the terminal velocity of particles in the seeding and feeding area is from -1 to -4 m/s, but most of them are less than -2.5 m/s. According to the cloud phase in Fig. 4, the particles are snowflakes in the cloud seeding and feeding area. The particle size is related to the shape of snowflakes and the final falling velocity, so it difficult to accurately quantify particle size. Table 1 lists the relationship between snowflake diameter and terminal velocity.

Table 1  the relations of cloud particles and diameter (Tao R., et al. TGRS, 2020)

| Study | Relation | Snow Type |
|---|---|---|
| Brandes (2002) [22] | $v_t = -0.1021 + 4.932D - 0.9551D^2 + 0.07934D^3 - 0.002362D^4$ | Rain |
| Jeong-Eun (2015) [23] | $v = 1.03D^{0.71}$ | Needle |
| Locatelli and Hobbs (1974) [24] | $v = 1.3D^{0.66}$ | Lump graupel |
| | $v = 0.79D^{0.27}$ | Densely rimed aggregates |
| | $v = 0.81D^{0.16}$ | Unrimed aggregates |
| This study | $v = 1.03D^{0.25}$ | low terminal velocity cases |
| | $v = 1.29D^{0.29}$ | high terminal velocity cases |

According to the speculation from table 1, the size of the snow particles in the cloud is distributed between 1mm and 6mm, and most of them are below 3mm.

 Yes, the sinking velocity is the particle terminal velocity. Thanks.

28. Lines 207-210: "During being seeded, ice particles were the main component in the cloud. After being seeded, the ice particles in the lower part of the feeding cloud lasted for a long time (maintaining the whole t3 period), while the supercooled water layer at the top was obvious": This needs more explanation.
 **Answer:**
"Before seeding, the larger downward mean Doppler velocity (Figure 2b) was detected in the lower part of the seeding cloud, which indicates that the cloud process has transformed from ice to snow with large particle sizes. Snowflakes, as seeders, fall into the mixed phase cloud containing supercooled water, so that the Wegener-Bergeron-Findeisen effect occurred. That effect causes the mixed phase cloud to rapidly transform into ice.  Because it takes time for particles to fall, so the seeding will continue to the middle and lower parts of the feeding clouds, and snow keeps for a long time (maintaining the entire t3 period); In the top region of the unaffected feeding cloud, the cloud phase remains supercooled water, which is consistent with the observation results in Shupe (2007)."  Thanks.

29. Lines 211-212: "The instantaneous water vapor flux structure (Fig. 4b) indicates that the seeding cloud is smaller than the feeding cloud": smaller in size or water vapor flux?
**Answer:**   the sentence has been rewritten as "From Figure 4, it can be seen that the instantaneous water vapor flux of the seeding cloud is smaller than that of the feeding cloud." Thanks.

30. "seeing-feeding" in Lines 213-214 should be "seeding-feeding";
**Answer:**  OK, thank you very much.

31. Lines 229-233: not clear;

**Answer** :   This paragraph has been revised as "The cloud particle with larger

diameter has a larger falling speed under the action of gravity. In order to reveal the relationship between particle size and echo signal in the process of seeder and feeder. The statistical classification method of equal samples is adopted to find the relationship. All signal values (echo reflectivity, radial velocity, spectral width,

particle falling velocity, and vertical airflow velocity) are reordered according to their corresponding echo reflectivity values from small to large, and then compared in the equal sample." Thanks.

32. Line 236: Change "Following to this principle" to "Following this principle";

**Answer**:OK, thank you very much.

33. Lines 240-241: "……the corresponding average profile of cloud particle parameter profile for the three intensity echoes is also gained" needs rewritten;

**Answer**:It has been rewritten as "…… the corresponding average profile of cloud particle parameter for the three intensity echoes is also obtained." Thanks.

34. Lines 287-288: "…… after the seeding, the cloud particle size distribution and particle velocity of the bilayer cloud reach a relatively balanced and stable state through complex thermodynamic and dynamic interactions": needs more physical explanation.

**Answer**:Due to the fact that echo reflectivity factor, radial velocity, and falling terminal velocity reflect particle size, and spectral width reflects particle size distribution and particle category. in the end of seeding , the cloud particle size distribution and particle velocity of the bilayer cloud may reach a relatively balanced and stable state through the complex microphysical and dynamic interactions in the t2 period." Thanks.

35. Lines 310-312: "Therefore, the Effective Seeding Depth (ESD) is defined as the height difference between the top height of the feeding cloud and from the height down to the height of the maximum correlation coefficient, which represents the influence of seeders on the seeding cloud": I do not understand.

**Answer**:We confused. The last three words are " the feeding cloud" in the sentence. In the revised manuscript the error has been modified. Thanks.

36. What does it mean by "sowing effect" in Line 315?

**Answer**:That is a clerical error. "sowing effect" has been changed into "seeding effect" in Line 315. Thanks.

37. There are more expression problems in the remaining part of the text. I suggest the quality of the whole text to be carefully checked and revised.

**Answer**:Thanks. We have checked and revised manuscript sentence by sentence to

avoid mistakes.

**Response to reports from Anonymous Referee #1**

**General comments:**

The manuscript shows the microphysical characteristics of cloud caused by seeder-feeder process, based on ground-based Ka-band millimeter-wave cloud radar (MMCR) and microwave radiometer (MWR) in Xi'an, China. With the definition of the height difference (HD) and the effective seeding depth (ESD), this study shows detailed features of the reflectivity and radial velocity for seeder-feeder phenomena, which

would deepen our understanding of the physical process. however , many phrases and

explanations are not clear . I think the quality of the manuscript needs more improvement.

**Response:** Thank you very much for your nice comments. Your question and suggestion are very helpful for us to improve the quality of our paper. We appreciate the reviewer's thoughtful review and constructive comments. The following is our point-to-point replies.

**Specific Comments:**
1. Since the definitions of HD and ESD are key parts in this paper, the authors are suggested to make them more clearly with physical concept, especially for ESD. The ESD defined in Line 310-313 is still confused and the role of autocorrelation between reflectivity factor and falling velocity is not clear.

**Answer**:

Thank you very much for your nice reminding.

HD is the height difference between the base height of seeding cloud (BHSC) and the top height of feeding cloud (THFC). BHSC is the average value of the seeder cloud base during the t1 period, and THFC is the average value of the feeder cloud top during the t1 period.

This definition has been reinterpreted and explained in the revised manuscript (Line 142-153).

ESD is the depth at which the feeding cloud is directly affected by seeding during t2 period, and is defined using the correlation coefficient between the particle terminal velocity and the reflectivity factor. The particle terminal velocity and reflectivity factor increase with the increase of particle size. As their correlation coefficient increases, it indicates that the particle size is constantly increasing due to the effect of seeding. When the correlation coefficient starts to decrease, it indicates that the particle size also starts to decrease, and the influence of seeding begins to weaken or transfer to other positions. Therefore, the position with the highest correlation coefficient is defined as the final position in the feeding cloud affected by seeding. the Effective Seeding Depth (ESD) is defined as the height difference between the top height of the feeding cloud and from the height down to the height of the maximum correlation coefficient, which represents the influence of seeders on the feeding cloud

during t2 period.

This definition has been reinterpreted and explained in the revised manuscript (Line 328-334).

2. It is found that the updraft flow is very important for seeder-feeder in this study. It is suggested to distinguish the properties of cloud, such as if it is convective or stratiform cloud. It may make the clusters' features more vivid.

**Answer:** Thank you very much for your nice reminding.

The updraft flow in convective clouds or stratiform clouds is different greatly and can be used to distinguish them. Our research focus on the "seeding-feeding" process between multi-layered stratiform clouds. In the future, we will study the "seeding-feeding" process in convective clouds.

3. How to determine the time of t1? The description in this paper is very subjective and needs to be given more objective indicators

**Answer:** t1 is the period from the moment when the stable double-layer cloud appear until the start of seeding. The description has been revised in the revised manuscript (Line 146).

4. How to use cloud radar power spectrum to separate air velocity and particle falling velocity? More detailed explanation should be given.

**Answer:**

The retrieval of vertical airflow velocity from cloud radar power spectrum is not easy.

In the identified ice particle region and mixed phase region of stratiform cloud, the turbulence inside the cloud is very small and can be ignored, and the first point on the left side of the power spectrum represents the information of the smallest particle detected by the radar. When particle size is small enough to ignore its terminal velocity, the first point of the power spectrum can be used to retrieve vertical airflow velocity, that is, the small particle tracer method. In this study, the echo intensity of -21 dBZ was used as the threshold for radar detection volume containing small particles. When the echo intensity was less than -21 dBZ, it can be considered that the particle size was small enough to be used as a tracer particle to retrieve vertical airflow velocity. Meanwhile, if the spectral width was less than 0.4m/s, it was considered that the turbulence was small and could be ignored. In the identified supercooled water region, the peak position of the liquid cloud particle is used to obtain the vertical velocity of airflow (Wei et al., 2019). When it drizzles, the *SP* of MMCR usually shows the bimodal distribution, and the vertical velocity of the airflow in the cloud can be obtained by the bimodal position of the liquid cloud particles (Wei et al., 2019; Luke et al., 2010 and 2013). The vertical radial velocity is the combination of the particle terminal velocity and the air vertical motion. Therefore, the cloud particle terminal velocity can be obtained by subtracting the vertical airflow velocity from the radial velocity.

The detailed explanation has been added in the revised manuscript (Line 127-140).

5. There is a strong correlation between the seeder-feeder mechanism and the cloud particles phase. The determination of particle phase states and the process of change are not given

**Answer:** Yes , the seeder and feeder process can cause changes in cloud particles

phase. The determination of particle phase states is important for the "seeder-feeder" study. In this manuscript, "Cloud particle phase identification adopts cluster analysis method (Shupe, 2007). The specific process takes cloud reflectivity factor, particle radial velocity and spectrum width measured by the MMCR, and atmospheric temperature measured by MWR as input parameters for cloud phase identification." which can be found in lines 120-123 of the manuscript.

6. As shown in Figure 4, the cloud is dominated by ice crystals and snow during t2, this situation does not trigger the seeder-feeder mechanism.

**Answer:**

Figures 2-4 both show that particles in the upper cloud begin to fall into the lower cloud from T0. The cloud phase states are snowflakes or ice crystals in the feeding cloud where particles are seeded during t2. There are still mixed phase clouds or supercooled water clouds in the feeding cloud where particles were not seeded during t2. This indicates that the large ice particles fall through the supercooled liquid layer, they can initiate the glaciation of the cloud layer through the WBF process and/or grow by riming. This phenomenon has also been observed in the Ref. (Fabiola Ramelli, 2020).

7. There are some grammar and expression mistakes in this paper, which need to be modified.

**Answer:** Thanks. We have checked and revised manuscript sentence by sentence to

avoid mistakes.

More detailed comments:
1. Line 1: macro or micro??
**Answer:** it is macro.

2. The past and present tense is mixed in some paragraphs, which needs to be uniformed. For example, Line 13 "shows", Line 19 "showed";
**Answer:** Thank you very much. We have checked and revised manuscript sentence

by sentence to avoid mistakes.

3.  Line 9: "Because" is unwanted;
**Answer:** Thanks.
    "Because" in Line 9 has been deleted in the revised manuscript.

4.  Line 13: "China Xi'an" is wrong expressed;
**Answer:** Thanks. It has been changed into "Xi'an, China".

5.  Line 67: "were focused on analysis" ?
**Answer:** This sentence has been deleted in the revised manuscript. Thanks.

6.  Line 92 shows that "but Fig. 3c still shows that after this time, cloud particles still sink (at 02:45 BJT, sinking velocity –0.5 ms$^{-1}$) on the feeding cloud top". Figure 6 gives the autocorrelation coefficient profile in t2 period and shows positive correlations. It's better to add some figures and discussions to explore the relationships between reflectivity factor and particle descent velocity in t3 period.
**Answer:** Thank you for this suggestion.
    The autocorrelation coefficient and correlation coefficient profiles in t3 period are also calculated and shown in the following figure I. This section has been added to Figures 6 (c) and 6 (d) in the revised version.

[Figure]

Figure I The autocorrelation coefficient profile (a) between cloud particle terminal velocity and reflectivity factor at each layer from top to bottom in the bilayer cloud in t3 period, and the correlation coefficient profile (b) between the average descent velocity of cloud particle in the HD region and reflectivity factor at different height layers in t3 period.

    The autocorrelation coefficient profile during the t3 period is shown in Figure 6c, which shows that as the height decreases from the middle of HD (~5 km) to the upper part of the feeding cloud (~3.1 km), the autocorrelation coefficient increases from 0.2 to 0.5, and the increase in echo reflectivity and correlation coefficient is smaller than that of t2 at the same height, which indicates that the particle size continues to

increase as the height decreases, while the impact of seeding on feeding clouds is limited due to the insufficient seeding. The echo reflectivity factor reaches its maximum between 1 and 2 km, but the correlation coefficient does not increase synchronously, but oscillates and decreases, indicating that the increase in echo reflectivity factor is not only caused by the increase of particle size, but also by the increase of particle number. From Figure 6d, there is no clear corresponding relationship between the non-autocorrelation coefficients and the reflectivity factor during the t3 period.

The above text has also been added to lines 372 to 380 of the revised version.

7. Line 189: should be "table 3".
**Answer:** Thanks. It has been changed into "table 3".

8. Line 518: "seeing-feeding"??
**Answer:** Sorry, it is a typo, and it should be "seeding-feeding".

9. Based on the observation data of the ground-based Ka-band millimeter-wave cloud radar (MMCR) and microwave radiometer (MWR) in spring and autumn from 2020 to 2022(line 12), Table 4 show the observation results by MMCR from winter to the next spring from 2021 to 2022, which one is correct?
**Answer:** it should be "in spring and autumn from 2021 to 2022(line 12)", and has been revised.

10. The author should give all the spelling about CFAD in Fig. 7 to Fig.11。

**Answer:** Thanks.
"CAFD" in the manuscript has been changed into "distribution of probability density (DPDH)", and it is more accurate. The all the spelling about DPDH has been added in Fig. 7 to Fig. 11.

11. It should illustrate the exact numbers to describe how these parameters changed in different height or regions (et al., THSC-BHSC, HD, THSC), rather than "varies significantly during seedings", "all increased correspondingly", "decreased significantly".

**Answer:**

Thank you very much for this suggestion, the exact numbers descriptions have been added in the revised manuscript, such as Lines 292-298 and Lines 311-312.

12. Line 482 is very confusing.
**Answer:** "……that the HD height is high, and the ESD thickness is thin during seeding process." has been changed into "The higher the height of HD, the thinner the

thickness of ESD during seeder-feeder process"

13. Line 388, "the f type I" may be "the type I"
**Answer:** Yes, "f" is a typo, and it has been deleted in the revised manuscript.

14. Numbers of horizontal axis in Fig. 7, Fig.9 and Fig. 11 are italic, which are different to other figures. There are some more blanks to fit all figures.
**Answer:** Thanks, Numbers of horizontal axis in Fig. 7, Fig.9 and Fig. 11 have been revised into Ortho. And all the figures in the revised manuscript have been updated.

**Author's changes in manuscript**

We have carefully revised the manuscript base on the opinions of reviewers. The modified part has been marked using track mode. The specific modification list is as follows:

1)Some new contents are added in the revised manuscript.

Based on the comments of Anonymous Referee #1, the correlation during periods t3 in the case study was analyzed, the contents have also been added in lines 372 to 380 of the revised version and two figures c and d in Figure 6 were added.

2)The phrases and explanations in the manuscript has been modified and corrected

3)Some specialized vocabularies have been corrected.

① "Rain flag" in table 4 has been changed to "virga".
② "Particle final falling velocity" has been changed to "particle terminal velocity" in the revised manuscript.
③ "the sowing effect" has been modified to "the seeding effect".
④ "the contour frequency by altitude diagram (CFAD) with height" has been modified to "the distribution of probability density with height (DPDH)".

4)The grammar and tense have been corrected.

The observation phenomena described all use the past tense.

5)the definitions of HD and ESD have been clarified and descripted.

① HD is the height difference between the base height of seeding cloud (BHSC) and the top height of feeding cloud (THFC). BHSC is the average value of the seeder cloud base during the t1 period, and THFC is the average value of the feeder cloud top during the t1 period.
② ESD is the depth at which the feeding cloud is directly affected by seeding during t2 period, and is defined using the correlation coefficient between the particle terminal velocity and the reflectivity factor. The particle terminal velocity and reflectivity factor increase with the increase of particle size. As their correlation coefficient increases, it indicates that the particle size is constantly increasing due to the effect of seeding. When the correlation coefficient starts to decrease, it indicates that the particle size also starts to decrease, and the influence of seeding begins to weaken or transfer to other positions. Therefore, the position with the highest correlation coefficient is defined as the final position in the feeding cloud affected by seeding. the Effective Seeding Depth (ESD) is defined as the height difference between the top

height of the feeding cloud and from the height down to the height of the maximum correlation coefficient, which represents the influence of seeders on the feeding cloud during t2 period.

This definition has been reinterpreted and explained in the revised manuscript (Line 328-334).

6)The format of references has been revised, and they are references 38, 43 and 49.

7) The figures size (Figs. 5-12) and proportion have been adjusted and unified.